# FEAST YOUR EYES: MIXTURE-OF-RESOLUTION ADAPTATION FOR MULTIMODAL LARGE LANGUAGE MODELS

**Gen Luo**[1,2]**, Yiyi Zhou**[1]**, Yuxin Zhang**[1]**, Xiawu Zheng**[1]**, Xiaoshuai Sun**[1]**, Rongrong Ji**[1][✉]

[1]Key Laboratory of Multimedia Trusted Perception and Efficient Computing,
  Ministry of Education of China, Xiamen University, 361005, P.R. China.
[2]OpenGVLab, Shanghai AI Laboratory.

## ABSTRACT

In existing multimodal large language models (MLLMs), image resolution plays a significant role for granular visual recognition. However, directly increasing image resolution leads to expensive computational cost for MLLMs. In this paper, we reveal that a combination of low- and high-resolution visual features can efficiently mitigate this shortcoming. Based on this principle, we propose a novel and efficient method for MLLMs, termed *Mixture-of-Resolution Adaptation* (MRA). In particular, MRA adopts two visual pathways for images of different resolutions, where high-resolution visual information is embedded into the low-resolution pathway via the novel *mixture-of-resolution adapters* (MR-Adapters). This design also greatly reduces the input sequence length of MLLMs. To validate MRA, we apply it to a recent MLLM called LLaVA, and term the new model *LLaVA-HR*. We conduct extensive experiments on 17 vision-language (VL) tasks, which show that LLaVA-HR outperforms existing MLLMs on 15 VL tasks, *e.g.,* +5.2% on TextVQA. More importantly, both training and inference of LLaVA-HR remain efficient with MRA, *e.g., **20 training hours** and **faster inference speed** than LLaVA-NeXT. Source codes are released at: LLaVA-HR.

## 1 INTRODUCTION

Driven by the remarkable success of large language models (LLMs) (Touvron et al., 2023; Chen et al., 2020), research on multi-modal large language models (MLLMs) also receives an influx of interest in both academia and industry (Liu et al., 2023b; Luo et al., 2023; Alayrac et al., 2022; Chen et al., 2022; 2023c). Numerous efforts have been recently devoted to extending LLMs to more modalities, achieving breakthroughs on various vision-language tasks (Goyal et al., 2017; Singh et al., 2019; Hudson & Manning, 2019). Despite their success, existing MLLMs still fall short of granular visual recognition. For instance, the powerful GPT4-V also suffers from visual hallucinations when identifying small and occluded objects (Tong et al., 2024). This shortcoming inevitably limits the practical use of MLLMs.

To compensate for this shortcoming, early practitioners often resort to scaling up model size and increasing per-training data size (Alayrac et al., 2022; Li et al., 2023b; Bai et al., 2023). For instance, InstructBLIP (Dai et al., 2023) adopts over 129M image-text pairs for vision-language (VL) alignments, showing that a larger visual encoder is beneficial for MLLMs. Similarly, Qwen-VL (Bai et al., 2023) also increases the parameters of visual encoder to 1.9 billion and uses 1.5 billion image-text pairs for pre-training. Despite effective, this paradigm is prohibitively expensive, which often consumes about thousands of GPU hours.

Orthogonal to these works, we study the visual shortcoming of MLLMs from the perspective of image resolutions. As revealed in previous VL research (Jiang et al., 2020; Tong et al., 2024), increasing the resolution of input images is a straightforward solution for visual recognition, which becomes more important for MLLMs that involve fine-grained visual reasoning (Rose et al., 2023). As shown

---

[✉]Corresponding author.

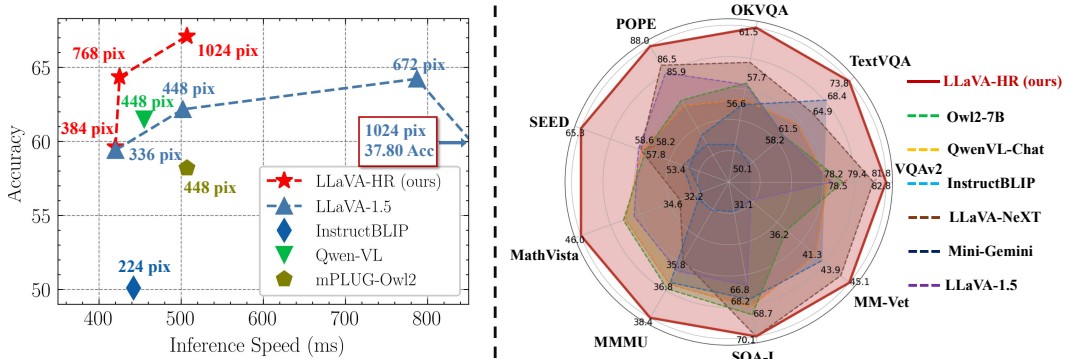

Figure 1: **Comparison between existing MLLMs and LLaVA-HR on TextVQA (left) and various benchmarks (right).** Increasing image resolution is effective yet expensive for fine-grained visual understanding. In contrast, LLaVA-HR can efficiently adapt high resolution to boost performance.

in Fig. 1, increasing the resolution of LLaVA-1.5 (Liu et al., 2023a) from $384 \times 384$ to $672 \times 672$ can bring obvious performance gains (+4.6%) on TextVQA (Singh et al., 2019). However, the use of high-resolution images will greatly exacerbate the already high computational cost of MLLMs. For instance, $448 \times 448$ resolution will increase the computation complexity of LLaVA by about 1.4 times compared with the default $336 \times 336$. In addition, the training will become unstable as the resolution is greatly increased[1], *e.g.*, a sharp drop at $1,022 \times 1,022$ resolution in Fig. 1. Although such an issue can be overcome by dividing high-resolution images into small patches via the dynamic slicing strategy Liu et al. (2024a), its computational cost still remains expensive for MLLMs.

In this paper, we focus on the efficient high-resolution image adaptation of MLLMs and propose a novel method called *mixture-of-resolution adaptation* (MRA). As shown in Fig. 2, MRA adopts an innovative dual visual pathway design to process the input images of high- and low-resolutions simultaneously. Specifically, one pathway aims to encode global information of low-resolution images, while the other one serves to capture fine-grained semantics from high-resolution images. Meanwhile, these two pathways are closely interacted via the novel *mixture-of-resolution adapters* (MR-Adapters), which embeds the high-resolution visual information into the low-resolution modeling. In this way, we can use a much fewer number of visual tokens to represent the input images from macro- to micro-views. With the careful design of dual-pathway structure, MRA can easily scale the image resolution up to $1,024 \times 1,024$ pixels while maintaining high efficiency.

To validate MRA, we apply it to a recent MLLM called LLaVA (Liu et al., 2023b;a), and term the new model as LLaVA-HR. We conduct extensive experiments on 17 vision-language (VL) tasks, including common VL tasks like VQAv2 (Goyal et al., 2017) and MLLM benchmarks such as POPE (Li et al., 2023c). Experimental results show that LLaVA-HR outperforms existing MLLMs on 15 of 17 VL tasks, *e.g.,* +9.6% over LLaVA-1.5 on TextVQA. More importantly, the training and inference of LLaVA-HR are cost-effective. In particular, the pre-training and instruction tuning of LLaVA-HR (7B, $1,024 \times 1,024$) only take a total of 20.7 hours on 8 A800 GPUs, which is **hundreds of times cheaper** than InstructBLIP (Dai et al., 2023) and Qwen-VL (Bai et al., 2023). Under the same high-resolution setting, its inference speed is **consistently faster** than LLaVA-1.5 (Liu et al., 2023a) and LLaVA-Next Liu et al. (2024a).

In summary, our contributions are three folds:

- We propose a novel and efficient adaptation scheme, termed *mixture-of-resolution adaption* (MRA), which adopts a novel dual visual pathway design to obtain the benefits of high-resolution visual information while keeping training and inference efficient.

- We propose a novel *mixture-of-resolution adapter* (MR-Adapter) for MRA, which can embed the high-resolution information into the low-resolution visual pathway to improve visual descriptive power.

- Based on MRA, we propose a powerful MLLM, coined LLaVA-HR, which outperforms existing MLLMs on 15 of 17 VL tasks and are much more efficient than most MLLMs.

---

[1]Visual encoders like CLIP-ViT are pre-trained with low resolution, and the significant increase of resolution may hurt feature representations.

## 2 RELATED WORK

### 2.1 MULTIMODAL LARGE LANGUAGE MODELS

Driven by the great successes of large language models (LLMs) (Gilardi et al., 2023; Touvron et al., 2023; Chen et al., 2020), growing interest has been aroused in building end-to-end multimodal large language models (MLLMs) (Liu et al., 2023b; Zhu et al., 2023; Luo et al., 2023; Bai et al., 2023; Fuyu-8B, 2023; Peng et al., 2023; Luo et al., 2024a;b). In particular, most existing MLLMs adopt a modular structure (Luo et al., 2023; Liu et al., 2023b), which utilizes an intermediate network to project the visual features into the word embedding space of the LLM. Then, the LLM is used to accomplish various VL tasks in an autoregressive manner. Based on the modular structure, existing MLLMs can be distinguished by the designs of the intermediate network. Popular MLLMs represented by LLaVA (Liu et al., 2023b) often adopt a linear projection layer or an MLP layer to connect the visual encoder and the LLM (Liu et al., 2023b; Chen et al., 2023a;c; Peng et al., 2023). The other works employ sampler-based modules to bridge the gap between the visual encoder and the LLM (Bai et al., 2023; Alayrac et al., 2022; Li et al., 2023b). These sampler-based modules can effectively reduce the number of visual tokens, but often requires a large-scale pre-training to achieve a promising performance (Bai et al., 2023; Li et al., 2023b). Despite the effectiveness, the low-resolution visual perception still limits the performance of existing MLLMs in fine-grained tasks.

### 2.2 HIGH-RESOLUTION MULTIMODAL LARGE LANGUAGE MODELS

To improve the perception ability of MLLMs, increasing attentions have been focused on high-resolution MLLMs (Liu et al., 2024a; Li et al., 2024c; Liu et al., 2024b; Li et al., 2024b; Chen et al., 2024b). Among them, most methods (Li et al., 2024c; Liu et al., 2024a) adopt the dynamic slicing strategy to divide a high-resolution image into multiple low-resolution patches. By doing so, pre-trained visual encoders can maintain their default resolutions for adapting high-resolution processing, and support images with flexible aspect ratio. For example, Monkey (Li et al., 2024c) and LLaVA-Next (Liu et al., 2024a) divide input images into a set of $448 \times 448$ patches for high-resolution visual understanding. Based on this framework, Chen et al. (2024b) and Dong et al. (2024) further explore the strategy to realize the optimal image division. Despite the effectiveness, their computational cost is still expensive as the image resolution increases. Orthogonal to these works, we aim to improve image resolution in an efficient way, which still lacks extensive explorations.

### 2.3 VISUAL REPRESENTATIONS FOR MULTIMODAL LARGE LANGUAGE MODELS

The pursuit of better visual representations has been a popular research trend in the VL community (Lu et al., 2019; Jiang et al., 2020; Radford et al., 2021). Early endeavors mainly explore the object-level features for VL models (Lu et al., 2019; Zhang et al., 2021). Driven by the large-scale image-text pre-training, grid features from CLIP (Radford et al., 2021) have demonstrated the great efficiency and generalization in MLLMs (Liu et al., 2023b; Chen et al., 2022; Alayrac et al., 2022). Based on grid features, existing researchers mainly improve visual representations by scaling up the visual encoder. For example, PaLI (Chen et al., 2022) increases the parameters of visual encoder to 3 billions and shows the significant performance boost of MLLMs. In contrast to these works, we improve the visual representations for MLLMs from the perspective of dual-branch network interactions, and propose a novel and efficient solution, namely mixture-of-resolution adaptation.

## 3 PRELIMINARY

We first recap the structure of multimodal large language models (MLLMs), which consists of an image encoder $\mathcal{F}_{\mathcal{I}}(\cdot)$, an intermediate network $\mathcal{F}_{\mathcal{P}}(\cdot)$ and an LLM $\mathcal{F}_{\mathcal{L}}(\cdot)$.

In particular, given an input image $I \in \mathbb{R}^{H \times W \times 3}$ and a textual instruction $T \in \mathbb{R}^L$, the visual tokens $\mathbf{F}_v \in \mathbb{R}^{(h \times w) \times d}$ are obtained via the image encoder, and the text tokens $f_t \in \mathbb{R}^{l \times d}$ are represented by the corresponding word embeddings. Based on the visual and textual tokens, the LLM will decode the target word step by step, formulated as

$$p_t = \prod_{s=1}^{S+1} \mathcal{F}_{\mathcal{L}}(R_s | \mathcal{F}_{\mathcal{P}}(\mathbf{F}_v), f_t, R_{0:s-1}). \tag{1}$$

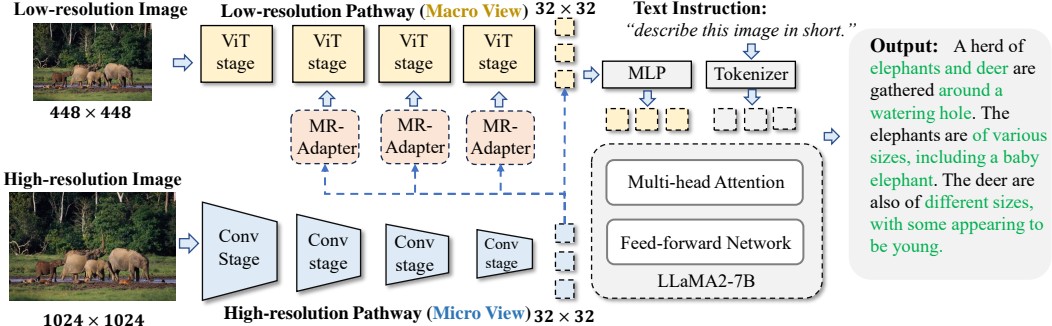

Figure 2: **Illustration of Mixture-of-Resolution Adaptation (MRA) and its deployment on LLaVA-HR.** MRA employs dual visual pathways to process high-resolution and low-resolution images, respectively. High-resolution information is embedded into the fast pathway via a novel mixture-of-resolution adapter (MR-Adapter).

Here, $p_t \in \mathbb{R}^m$ denotes the probabilities of the predicted word and $m$ is the size of word vocabulary.

In some MLLMs (Liu et al., 2023b;a), $\mathcal{F}_{\mathcal{P}}(\cdot)$ is often a stack of simple linear layers, which are used to directly project the visual tokens onto the semantic space of LLMs. Although simple and effective, this strategy inevitably leads to a longer visual sequence as the resolution increases, *e.g.,* 5,329 tokens for $1,022 \times 1,022$ resolution in LLaVA-1.5. In practice, processing such a large number of tokens is computationally expensive in MLLMs. To further reduce the number of visual tokens, recent advances adopt the sampler-based module for $\mathcal{F}_{\mathcal{P}}(\cdot)$, *e.g., QFormer* (Li et al., 2023b), which aggregates visual features into several query tokens that LLM can directly handle. Nevertheless, these methods often require large-scale pre-training to achieve VL alignments (Bai et al., 2023).

Based on the above analyses, we conclude that the main difficulty of high-resolution image adaptation lies in the rapidly growing visual sequence. This issue motivates us to further explore how to efficiently encode richer visual information with fewer visual tokens.

## 4 MIXTURE-OF-RESOLUTION ADAPTATION

### 4.1 OVERVIEW

To address the above issues, we propose a novel and efficient method for MLLMs, termed *mixture-of-resolution adaptation* (MRA). As shown in Fig. 2, MRA aims to embed high-resolution information into the low-resolution one via a dual pathway design. In this case, MRA can keep a smaller number of visual tokens while encoding richer visual information.

In particular, given the input images of two resolutions $I_l \in \mathbb{R}^{H_l \times W_l \times 3}$ and $I_h \in \mathbb{R}^{H_h \times W_h \times 3}$, the process of MRA can be formulated as

$$
\begin{aligned}
\mathbf{F}_v &= \mathcal{F}_{\mathcal{I}_l}\left(I_l, \mathcal{F}_{\mathcal{A}}\left(\mathbf{F}_{vh}; \theta_{\mathcal{A}}\right); \theta_{\mathcal{I}_l}\right), \\
\text{where} \quad \mathbf{F}_{vh} &= \mathcal{F}_{\mathcal{I}_h}(I_h; \theta_{\mathcal{I}_h}).
\end{aligned}
\tag{2}
$$

Here, $\mathbf{F}_{vh} \in \mathbb{R}^{h_h \times w_h \times d_h}$ and $\mathbf{F}_v \in \mathbb{R}^{h \times w \times d}$ denote the high-resolution features and the final visual features, respectively. And $\mathcal{F}_{\mathcal{I}_l}(\cdot)$ and $\mathcal{F}_{\mathcal{I}_h}(\cdot)$ are the visual encoders for high-resolution and low-resolution images, respectively. $\mathcal{F}_{\mathcal{A}}$ denotes the *mixture-of-resolution adapter* (MR-Adapter). Based on Eq. 2, the obtained visual features will be further processed by the LLM based on Eq. 1.

### 4.2 DUAL VISUAL PATHWAYS

As shown in Fig. 2, dual visual pathways, *i.e.,* $\mathcal{F}_{\mathcal{I}_l}(\cdot)$ and $\mathcal{F}_{\mathcal{I}_h}(\cdot)$ are the key design of MRA. To maximize their benefits, we consider the heterogeneous dual-branch design from two aspects.

**Visual functionality.** Firstly, the dual visual pathways process images from macro- and micro-views, which is inspired by the visual system of human being (Merigan & Maunsell, 1993; Robertson & Lamb, 1991). Particularly, Robertson & Lamb (1991) find that the visual system processes local

and global semantics via different pathways. Similar mechanisms in computer vision are not new. Previous works (Chen et al., 2021; Peng et al., 2021) like CrossViT (Chen et al., 2021) typically incorporate this feature into their network design for image classification.

However, the exploration of dual visual pathways in high-resolution adaptation for MLLMs can still bring new insights beyond previous works, *i.e.,* fewer visual tokens can also result in stronger visual understanding. Specifically, one visual pathway aims to capture fine-grained semantics from high-resolution images *i.e.*, processing images from local view. The other pathway is designed to encode global information from low-resolution images for a larger receptive field. In this case, MRA can not only efficiently process high-resolution images, but also greatly benefits from two complementary visual semantics.

**Visual alignment.** The alignment of two pathways is also challenging in MLLMs, which typically requires additional fusion layers like cross-attentions (Vaswani et al., 2017). Due to different resolutions, these two pathways often produce visual features of different shapes, impeding their quick alignments (Yu et al., 2019). To overcome this limitation, we adopt different downsampling rates for the low- and high-resolution pathways, respectively. Thus, their output features can keep the same spatial shape.

Based on the above motivations, $\mathcal{F}_{\mathcal{I}_l}(\cdot)$ and $\mathcal{F}_{\mathcal{I}_h}(\cdot)$ are designed as a vision transformer (ViT) (Dosovitskiy et al., 2020) and a convolutional network (CNN) (Liu et al., 2022), respectively. Specifically, CNN is equipped with a downsampling stride of 32 to process high-resolution images. ViT encodes low-resolution images with a downsampling stride of 14. Notably, such designs also ensure the efficiency of MLLMs, where the high-resolution images are processed by the efficient CNN, and the number of visual tokens is also kept small via the large downsampling stride.

## 4.3 MIXTURE-OF-RESOLUTION ADAPTER

To better collaborate the feature learning of two pathways, we propose a *mixture-of-resolution adapter* (MR-Adapter) to embed high-resolution information of CNN into different stages of ViT. This early fusion strategy can leverage ViT's deep Transformer layers to excavate fine-grained context from different visual sources.

In particular, given the visual features $\mathbf{F}_{vh} \in \mathbb{R}^{h \times w \times d_h}$ of the a high-resolution image, we embed them into the low-resolution visual pathway by

$$\mathbf{F}_{vl}^{i'} = \mathcal{F}_l(\mathbf{F}_{vl}^i; \theta_l) + g \cdot \mathcal{F}_h(\mathbf{F}_{vh}; \theta_h). \quad (3)$$

Here, $\mathbf{F}_{vl}^i \in \mathbb{R}^{h \times w \times d_l}$ are features from the i-*th* stage of ViT. $\mathcal{F}_l(\cdot)$ is a lightweight convolution layer with a residual connection. $\mathcal{F}_h(\cdot)$ denotes an MLP layer. $g$ is a dynamic score to control the weights of high-resolution information, defined by

$$g = \delta(W_2 \sigma(W_1 f_v)). \quad (4)$$

Here, $f_v \in \mathbb{R}^{2d}$ is the global average pooling of visual features $[\mathcal{F}_l(\mathbf{F}_{vl}^i), \mathcal{F}_h(\mathbf{F}_{vh})]$, where $[\cdot]$ denotes the concatenation operation. $W_1 \in \mathbb{R}^{2d \times \frac{d}{2}}$ and $W_2 \in \mathbb{R}^{\frac{d}{2} \times d}$ are two projection matrices. $\sigma$ and $\delta$ denote the activation function of *GELU* and *Tanh*, respectively.

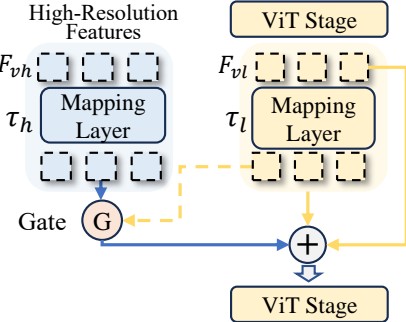

Figure 3: **Illustration of MR-Adapter.** MR-Adapter can dynamically embed the high-resolution features into the low-resolution pathway.

As shown in Fig. 2, high-resolution information can be fused with the features in each block of ViT. In this case, the low-resolution features of ViT also contain rich semantics, improving the visual descriptive power of MLLMs.

## 4.4 THE DEPLOYMENT ON MLLM

We apply MRA to LLaVA-1.5 (Liu et al., 2023a) and construct a new model, namely LLaVA-HR. Its training consists of two stages, *i.e.*, low-resolution pre-training and high-resolution instruction tuning.

**Stage 1: Low-resolution pre-training.** Similar to LLaVA (Liu et al., 2023b) and LLaVA-1.5 (Liu et al., 2023a), this stage aims to optimize the projector to align the visual features with the word embedding space of LLM. Therefore, the image encoder and the LLM are frozen during pre-training. Besides, we adopt low resolutions for two pathways, *i.e.,* $384 \times 384$ and $336 \times 336$. In this stage, the MR-Adapter is not inserted, and output features of dual pathways are upsampled to the same size and directly combined.

**Stage 2: High-resolution instruction tuning.** During instruction tuning, we increase the resolution of the high-resolution pathway, *e.g.,* from $384 \times 384$ to $1,024 \times 1,024$. And the low-resolution one is also accordingly adjusted to ensure the visual alignment of two pathways, *e.g.,* from $336 \times 336$ to $448 \times 448$. Meanwhile, the MR-Adapter is then applied to connect two visual pathways. Different from the first training stage, the entire MLLM will be fully optimized to better accommodate high-resolution images.

## 5 EXPERIMENTS

### 5.1 EVALUATIONS AND METRICS

**Multimodal benchmarks for MLLM.** We evaluate LLaVA-HR on six emerging multimodal benchmarks for MLLMs, including MME (Fu et al., 2023), POPE (Li et al., 2023c), SEED (Li et al., 2023a), MM-VET (Yu et al., 2023b), MMMU (Yue et al., 2023) and MathVista (Lu et al., 2023). In particular, MME and MM-VET evaluate the multimodal perception and cognition abilities of MLLMs. SEED extends the modalities of evaluation to images and videos. POPE aims to evaluate the visual hallucinations of MLLMs. MMMU and MathVista aim to evaluate the multi-discipline and math understanding ability, respectively. The metrics used in our paper follow their default settings.

**General visual question answering benchmarks.** We also evaluate LLaVA-HR on seven VL datasets, including VQAv2 (Goyal et al., 2017), GQA (Hudson & Manning, 2019), OKVQA (Marino et al., 2019), OCRVQA (Mishra et al., 2019), ScienceQA (Lu et al., 2022a), VizWiz (Gurari et al., 2018) and TextVQA. In particular, ScienceQA (Lu et al., 2022a) and VizWiz (Gurari et al., 2018) are two zero-shot tasks, and their samples are not appeared in our training data. We report the accuracy on the *test* set of OCRVQA, the *test* set of VizWiz, and the *val* set of OKVQA. We organize samples of these tasks in instruction formats of LLaVA-1.5 (Liu et al., 2023a).

**OCR-related benchmarks.** To validate the fine-grained recognition ability of LLaVA-HR, we further evaluate it on five text-rich image understanding tasks, including TextVQA (Singh et al., 2019), DocVQA (Mathew et al., 2021), InfoVQA (Mathew et al., 2022), AI2D (Kembhavi et al., 2016) and ChartVQA (Masry et al., 2022). For DocVQA and InfoVQA, we use the metric of ANLS. For remaining benchmarks, we use the accuracy as the metric. Results of LLaVA-HR on OCR-related benchmarks are evaluated by the VLMEvalKit Duan et al. (2024).

### 5.2 IMPLEMENTATION DETAILS

In LLaVA-HR, we use CLIP-ViT-L (Radford et al., 2021; Ilharco et al., 2021) and CLIP-ConvNeXt-L (Liu et al., 2022) as the dual visual paths to encode low- and high-resolution images, respectively. In LLaVA-HR-X, the CLIP-ConvNeXt-L is replaced with the stronger CLIP-ConvNeXt-XXL. The MR-Adapter is applied into the last three stages of ViT. Following LLaVA-1.5, we first pre-train LLaVA-HR on LCS-558K (Liu et al., 2023b), which contains $558k$ image-text pairs. During the pre-training stage, both the visual encoder and the LLM are frozen, and only the MLP projector is fine-tuned. AdamW (Kingma & Ba, 2014) is used as the optimizer, and the learning rate and batch size are set to 1e-3 and 256, respectively. Visual resolutions are set to $336 \times 336$ and $384 \times 384$ for the ViT and the CNN, respectively. During instruction tuning, we follow LLaVA-1.5 to use $665k$ VL instruction data. When fairly comparing with recent MLLMs like MM1 (McKinzie et al., 2024), we use additional 1.6M instruction data including ShareGPT4V (Chen et al., 2023b), LAION-GPT-4V (laion, 2023), ALLAVA (Chen et al., 2024a), LIMA (Zhou et al., 2024), OpenAssistant2 (Köpf et al., 2024), Tabmwp (Lu et al., 2022b), MathQA (Yu et al., 2023a), KVQA (Shah et al., 2019), Geometry (Lu et al., 2021), STVQA (Biten et al., 2019), ChartQA (Masry et al., 2022), DVQA (Kafle et al., 2018), AI2D (Kembhavi et al., 2016), LLaVA-Med (Li et al., 2024a), InfoVQA (Mathew et al., 2022) and MathV360k Shi et al. (2024). At this stage, the entire model is updated with a learning

Table 1: **Performance and efficiency comparisons of existing high-resolution adaptation solutions.** All experiments are conducted based on LLaVA-1.5. The training and inference costs are measured on NVIDIA A800s. "*Res.*" and '*V-Token*" denote image resolutions and the number of visual tokens, respectively. "*t/s*" denotes the number of generated tokens per second. "*N/A*" means that GPU memory overflows, so we reduce the batch size.

| Methods | Res. | V-Token | Vision-Language Tasks | | | | Training | GPU | Inference |
| | | | VQAv2 | TVQA | MME | POPE | Time ↓ | Memory ↓ | Speed ↑ |
|---|---|---|---|---|---|---|---|---|---|
| LLaVA-1.5 (Liu et al., 2023a) | 336 pix | 576 | 80.4 | 59.4 | 1461 | 86.2 | 15.6h | 28G | 23.8 t/s |
| +Resize | 448 pix | 1024 | 81.1 | 62.1 | 1493 | 87.2 | 19.4h | 49G | 19.9 t/s |
| +Resize | 672 pix | 2304 | 81.5 | 64.2 | 1498 | 87.9 | 31.8h | 79G | 12.7 t/s |
| +Resize | 1022 pix | 5329 | 74.2 | 37.8 | 1266 | 84.4 | 69.4h | N/A | 5.6 t/s |
| +Avg. Pooling | 756 pix | 729 | 80.6 | 59.6 | 1480 | 86.5 | 37.3h | 45G | 23.9 t/s |
| +CNN Encoder (Liu et al., 2022) | 768 pix | 576 | 80.3 | 64.6 | 1415 | 86.6 | 17.6h | 37G | 23.7 t/s |
| +Resampler (Jaegle et al., 2021) | 756 pix | 64 | 79.8 | 58.9 | 1403 | 85.8 | 36.5h | 40G | 27.6 t/s |
| +AnyRes (Liu et al., 2024a) | ∼1088 pix | ∼2880 | 81.7 | 65.1 | 1487 | 87.7 | 33.5h | 65G | 14.8 t/s |
| +MRA (ours) | 768 pix | 576 | 81.8 | 64.3 | 1524 | 88.0 | 18.2h | 38G | 23.5 t/s |
| +MRA (ours) | 1024 pix | 1024 | 81.9 | 67.1 | 1554 | 87.6 | 20.7h | 40G | 19.7 t/s |

rate of 2e-5. Besides, we increase the resolution of ViT and CNN to 448×448 and 1,024×1,024, respectively. The training epoch is set to 1 for pre-training and instruction tuning.

## 5.3 EXPERIMENTAL RESULTS

### 5.3.1 QUANTITATIVE ANALYSIS

**Comparison with high-resolution baselines.** In Tab. 1, we compare the performance and efficiency of MRA and existing high-resolution solutions on LLaVA-1.5 (Liu et al., 2023a). In this table, "Resize" aims to directly increase the image resolution. 'CNN Encoder" replaces the visual backbone with ConvNeXt (Liu et al., 2022), which uses a larger downsampling rate to reduce the number of visual tokens. "Avg. Pooling" and "Resampler" refer to the two pooling strategies for reducing the number of visual tokens. For "Resampler", we follow QwenVL-Chat and reduce the number of visual tokens to 64. "AnyRes" divides a high-resolution image into several sub-images (Liu et al., 2024a). From this table, we observe that directly increasing image resolution obviously improves the performance of two models on four tasks, *e.g.,* +4.8% of LLaVA-1.5 on TextVQA. However, the performance of LLaVA-1.5 drops significantly at the resolution of 1,024×1,024. To explain, the number of visual tokens greatly exceeds the pre-trained context length of the LLM, which easily causes the instability during training. Besides, we can also see that although several baselines can well maintain the inference efficiency, their benefits to performance are not obvious.

In particular, "Resampler" even hurts the model performance on four benchmark datasets, which often requires large-scale pre-training to achieve a promising performance. In contrast, as the most popular solution in existing literature (Liu et al., 2024a; Gao et al., 2024), "AnyRes" can effectively bring obvious performance gains on TextVQA and POPE. Nevertheless, the number of visual token increases

Table 2: **Ablation Study of MRA on LLaVA-1.5.** "Tune vision" means that the image encoder is fine-tuned.

| Methods | VQAv2 | TVQA | MME | POPE |
|---|---|---|---|---|
| LLaVA-1.5 (Liu et al., 2023a) | 78.5 | 58.2 | 1510.7 | 85.9 |
| +Tune vision | 80.4 +0.9 | 59.4 +1.2 | 1461.2 -49.5 | 86.2 +0.3 |
| +Dual-pathway | 81.3 +1.8 | 62.8 +4.6 | 1513.1 +2.4 | 87.2 +1.3 |
| +MR-Adapter | 81.8 +2.3 | 64.4 +6.2 | 1524.8 +14.1 | 88.0 +2.1 |
| +1024 resolution | 81.9 +2.4 | 67.1 +8.9 | 1554.9 +44.2 | 87.6 +1.7 |
| +13B LLM | 82.3 +2.8 | 68.1 +9.9 | 1540.9 +30.2 | 87.8 +1.9 |
| +1B Vision | 82.6 +3.1 | 70.9 +12.7 | 1487.3 -23.4 | 88.0 +2.1 |

significantly, leading to extremely high computational complexity. Compared to these methods, the performance of MRA is consistently improved from 768 × 768 resolution to 1,024 × 1,024 resolution. Besides, the total gain of MRA is more obvious than that of all compared methods, *e.g.,* +2.0% against AnyRes (Liu et al., 2024a) on TextVQA.

In addition to performance, the expenditure of LLaVA-HR is also cost-effective. In particular, increasing resolution from 336 × 336 to 1,022 × 1,022 slows down the training and inference of

Table 4: **Comparison with existing methods on four MLLM benchmarks.** "Param.", "Res." and "Data" refer to the parameters, the resolution and the training data, respectively. "t/s" refers to tokens per second. CogVLM-Chat and InternVL-1.2 use more data and parameters, so we mark it in gray.

| Method | Settings | | | General MLLM Benchmarks | | | | | | | Inference |
|---|---|---|---|---|---|---|---|---|---|---|---|
| | Param. | Res. | Data | MME | POPE | SEED | SEED$^I$ | MM-Vet | MMMU | MathVista | Speed |
| BLIP-2 (Li et al., 2023b) | 14B | 224 | 129M | 1293.8 | 85.3 | 46.4 | 49.7 | 22.4 | - | - | - |
| InstructBLIP (Dai et al., 2023) | 14B | 224 | 130M | 1212.8 | 78.9 | - | - | 25.6 | - | - | - |
| QwenVL-Chat (Bai et al., 2023) | 10B | 448 | 1.4B | 1487.5 | - | 58.2 | 65.4 | - | 35.9 | - | 17.0 t/s |
| Fuyu-8B (Fuyu-8B, 2023) | 8B | 600 | - | 728.6 | 74.1 | - | - | 21.4 | - | - | 15.6 t/s |
| mPLUG-Owl2 (Ye et al., 2023) | 8B | 448 | 400M | 1450.2 | - | 57.8 | - | 36.2 | 32.7 | - | 19.6 t/s |
| I-MoF (Tong et al., 2024) | 13B | 336 | 1.2M | - | 86.7 | - | - | 34.6 | - | - | - |
| LLaVA-1.5 (Liu et al., 2023a) | 7B | 336 | 1.2M | 1510.7 | 85.9 | 58.6 | 66.1 | 30.5 | - | - | 23.8 t/s |
| LLaVA-1.5 (Liu et al., 2023a) | 13B | 336 | 1.2M | 1531.3 | 85.9 | 61.6 | 68.2 | 35.4 | 36.4 | 27.6 | 16.1 t/s |
| LLaVA-HR | 7B | 1024 | 1.2M | **1554.9** | 87.6 | 64.2 | 70.6 | 31.5 | 35.2 | **28.5** | 19.7 t/s |
| LLaVA-HR | 13B | 1024 | 1.2M | 1540.9 | 87.8 | 64.5 | 70.9 | 35.5 | 35.7 | 27.7 | 15.0 t/s |
| LLaVA-HR-X | 14B | 1024 | 1.2M | 1487.3 | **88.0** | **65.3** | **71.4** | **40.3** | **36.6** | 28.1 | 12.9 t/s |
| *More Instruction Data:* | | | | | | | | | | | |
| LLaVA-NeXT (Liu et al., 2024a) | 7B | 1344 | 1.6M | 1519.0 | 86.5 | - | 70.2 | 43.9 | 35.8 | 34.6 | 14.8 t/s |
| SPHINX-intern2 (Gao et al., 2024) | 7B | 448 | 16M | 1260.4 | 86.9 | - | - | 36.5 | - | 35.5 | - |
| InternLM-XC (Zhang et al., 2023) | 7B | 224 | 1.1B | 1528.4 | - | - | - | 35.2 | - | 29.5 | - |
| Mini-Gemini (Li et al., 2024b) | 7B | 672 | 2.7M | **1546.0** | - | - | - | 41.3 | 36.8 | 32.2 | 16.2 t/s |
| MM1 (McKinzie et al., 2024) | 7B | 1792 | 1B | 1529.3 | 86.6 | 64.0 | 69.9 | 42.1 | 37.0 | 35.9 | - |
| CogVLM-Chat (Wang et al., 2023) | 17B | 490 | 1.5B | - | - | - | - | 51.1 | 41.1 | 34.5 | 11.5 t/s |
| InternVL-1.2 (Chen et al., 2023d) | 40B | 448 | 450M | 1687.0 | - | - | - | 48.9 | 51.6 | 47.7 | 11.3 t/s |
| LLaVA-HR† | 7B | 1024 | 2.7M | 1490.5 | **86.9** | **64.9** | **71.9** | **45.1** | **38.4** | **46.0** | 19.7 t/s |

LLaVA-1.5 by 344.8% and 325%, respectively. However, these costs are reduced to only 17.6% and 20.8% in LLaVA-HR. Despite better performance, the training and inference speeds of LLaVA-HR are three times faster than LLaVA-1.5. Besides, the costs of GPU memory also remain cheap for LLaVA-HR. For example, adapting the resolution of $1,024 \times 1,024$ for LLaVA-HR only consumes 40G GPU memory, but the same settings for LLaVA-1.5 will cause GPU memory overflow. These results greatly confirm the efficiency of our MRA and LLaVA-HR.

**Ablation studies.** In Tab. 2 and 3, we conduct comprehensive ablation studies for MRA on four benchmarks. Firstly, we validate each design of our MRA in Tab. 2. From these results, we find that each component obviously contributes to the final performance. For example, the dual visual pathways and the MR-Adapter provide +3.4% and +1.6% performance gains on TextVQA, respectively. After increasing the resolution to $1,024 \times 1,024$, the performance on TextVQA further boosts by +2.7%. In the second block of Tab. 2, we also ablate the parameter scale of the LLM and the visual encoder. Experimental results show that larger visual backbone or LLM will consistently improve the model performance, further confirming the scalability of MRA.

In Tab 3, we compare different designs in MRA. From these results, we find that a larger high-resolution visual encoder typically brings more gains than a larger low-

Table 3: **Different choices of MRA on LLaVA-HR.** "L-Res Path.", "H-Res Path." and "Fusion Direct." denote the low-resolution pathway, the high-resolution pathway and the fusion direction, respectively. Our final setting is colored in gray.

| Settings | Choices | VQAv2 | TVQA | MME | POPE |
|---|---|---|---|---|---|
| L-Res Path. | ViT-L | 81.8 | 64.4 | 1524.8 | 88.0 |
| | ViT-G | 81.7 | 65.3 | 1469.7 | 87.9 |
| H-Res Path. | ConvXt-L | 81.8 | 64.4 | 1524.8 | 88.0 |
| | ConvXt-XXL | 82.3 | 66.5 | 1479.2 | 87.9 |
| Fusion Direct. | High to Low | 81.8 | 64.4 | 1524.8 | 88.0 |
| | Low to High | 81.0 | 62.8 | 1463.5 | 87.3 |
| Insert Position | last 3 stages | 81.8 | 64.4 | 1524.8 | 88.0 |
| | last stage | 81.3 | 62.8 | 1513.1 | 87.2 |
| | last 2 stages | 81.6 | 63.8 | 1508.4 | 87.5 |
| | last 4 stages | 81.4 | 63.1 | 1461.6 | 87.5 |

resolution one. Besides, the fusion direction of MRA is also significant. Specifically, changing the fusion direction obviously degenerates the performance, *e.g.,* -61.3 on MME. Such results also confirm our design principle of MRA, *i.e.,* embedding high-resolution information in to low-resolution pathway. Meanwhile, the best choice of the insert position of MRA is the last 3 stages of ViT. These ablations further confirm the designs of MR-Adapter.

Table 5: **Comparison with existing methods on seven general visual question answering tasks.** SQA$^I$ refers to the *IMG* subset of ScienceQA.

| Method | Settings | | | General Visual Question Answering | | | | | | | Infer. |
|---|---|---|---|---|---|---|---|---|---|---|---|
| | Param. | Res. | Data | VQAv2 | GQA | OKVQA | OCRVQA | SQA$^I$ | VizWiz | TVQA | Speed |
| BLIP-2 (Li et al., 2023b) | 14B | 224 | 129M | 41.0 | 41.0 | 45.9 | 40.6 | 61.0 | 19.6 | 42.5 | - |
| InstructBLIP (Dai et al., 2023) | 14B | 224 | 130M | - | 49.5 | - | 44.8 | 63.1 | 33.4 | 50.7 | - |
| Shikra (Chen et al., 2023a) | 13B | 224 | 6.1M | 77.4 | - | - | - | - | - | - | - |
| IDEFICS-9B (IDEFICS, 2023) | 9B | 224 | 354M | 50.9 | - | 38.4 | - | - | 35.5 | 25.9 | 30.5 t/s |
| IDEFICS-80B (IDEFICS, 2023) | 80B | 224 | 354M | 60.0 | - | 45.2 | - | - | 36.0 | 30.9 | - |
| QwenVL-Chat (Bai et al., 2023) | 10B | 448 | 1.4B | 78.2 | 57.5 | 56.6 | **70.5** | 68.2 | 38.9 | 61.5 | 17.0 t/s |
| Fuyu-8B (Fuyu-8B, 2023) | 8B | 600 | - | 74.2 | - | 60.6 | - | - | - | - | 15.6 t/s |
| mPLUG-Owl2 (Ye et al., 2023) | 8B | 448 | 400M | 79.4 | 56.1 | 57.7 | - | 68.7 | 54.5 | 58.2 | 19.6 t/s |
| I-MoF (Tong et al., 2024) | 13B | 336 | 1.2M | 79.3 | - | - | - | - | - | 58.7 | - |
| LLaVA-1.5 (Liu et al., 2023a) | 7B | 336 | 1.2M | 78.5 | 62.0 | - | - | 66.8 | 50.0 | 58.2 | 23.8 t/s |
| LLaVA-1.5 (Liu et al., 2023a) | 13B | 336 | 1.2M | 80.0 | 63.3 | - | - | **71.6** | 53.6 | 61.3 | 16.1 t/s |
| LLaVA-HR | 7B | 1024 | 1.2M | 81.9 | 64.2 | 58.9 | 68.4 | 67.9 | 48.7 | 67.1 | 19.7 t/s |
| LLaVA-HR | 13B | 1024 | 1.2M | 82.3 | 64.8 | 60.7 | 67.7 | 70.1 | **57.9** | 68.1 | 15.0 t/s |
| LLaVA-HR-X | 14B | 1024 | 1.2M | **82.6** | **65.2** | **61.5** | 69.0 | 69.7 | 56.6 | **70.9** | 12.9 t/s |

Table 6: **Comparison with existing MLLMs on five multimodal OCR-related benchmarks.**

| Method | Param. | Res. | Data. | TextVQA | DocVQA | InfoVQA | AI2D | ChartQA |
|---|---|---|---|---|---|---|---|---|
| QwenVL (Bai et al., 2023) | 10B | 336 | 1.4B | 63.8 | 65.1 | 35.4 | - | 65.7 |
| Monkey (Li et al., 2024c) | 10B | 1344 | 1.4M | 67.6 | 66.5 | 36.1 | 62.6 | - |
| LLaVA-NeXt (Liu et al., 2024a) | 7B | 1344 | 1.6M | 64.9 | - | - | 66.6 | 54.8 |
| TextMonkey (Liu et al., 2024b) | 10B | 1344 | 2.5M | 65.9 | 73.0 | 28.6 | - | 65.5 |
| DocOwl-1.5-Chat (Hu et al., 2024) | 8B | 4032 | 4M | 68.6 | 82.2 | 50.7 | - | 70.2 |
| CogAgent Hong et al. (2023) | 18B | 1120 | >300M | **76.1** | 81.6 | 44.5 | - | 68.4 |
| LLaVA-HR† | 7B | 1024 | 2.7M | 73.8 | **85.8** | **52.3** | **75.3** | **77.6** |

**Comparison with existing MLLMs.** In Tab. 4 and 5, we compare LLaVA-HR with existing MLLMs on 13 VL tasks. On the six MLLM benchmarks, we observe comprehensive advantages of LLaVA-HR against existing MLLMs. In particular, LLaVA-HR achieves 1554.9 scores in MME benchmark, outperforming LLaVA-1.5 by +23.6. On POPE, a benchmark including video evaluations, LLaVA-HR-X still outperforms existing MLLMs by a large margin, *i.e.,* +3.7% gains. Besides, LLaVA-HR achieves the best performance on the benchmark for visual hallucinations, *i.e.,* POPE, suggesting that its visual hallucinations are greatly alleviated. Meanwhile, we also compare the recently proposed MLLMs in the second block of Tab. 4. In particular, we still observe the better performance of LLaVA-HR against LLaVA-NeXT (Liu et al., 2024a), SPHINX-intern2 (Gao et al., 2024), Mini-Gemini (Li et al., 2024b) and MM1 (McKinzie et al., 2024), *e.g.,* +3.0% on MM-Vet.

Tab. 5 gives the performance comparison on common VL tasks. On in-domain tasks, LLaVA-HR achieves the best results on three tasks, *e.g.,* 82.6 on VQAv2 and 61.5 on OKVQA. On OCRVQA, Qwen-VL-Chat collects more in-domain data for training, so it performs better than LLaVA-HR. Under the zero-shot setting, we can observe more significant advantages of LLaVA-HR on the fine-grained tasks, *e.g.,* VizWiz. Most notably, even Qwen-VL-Chat is pre-trained with 24.8M OCR samples, it still performs worse than LLaVA-HR-X on TextVQA. These results suggest the significance of high resolution for these tasks. In contrast, most images of ScienceQA are synthetic and of low resolution, so the advantages of LLaVA-HR are not obvious. Overall, these results greatly confirm the effectiveness and generalization of LLaVA-HR and our MRA.

Tab. 6 compares LLaVA-HR and existing MLLMs on text-rich image understanding tasks. Compared to common MLLM benchmarks and VQA benchmarks, these OCR-related benchmarks pose a higher requirement for image resolution. As shown in Tab. 6, low-resolution MLLMs like QwenVL often perform inferior to high-resolution ones, *e.g.,* -4.8% on TextVQA compared to DocOwl-1.5-Chat Hu et al. (2024). However, we still observe that LLaVA-HR greatly outperforms existing MLLMs on five benchmarks. For example, although DocOwl-1.5-Chat has larger model size, input resolution and data size, LLaVA-HR also demonstrates superior fine-grained text recognition ability, *e.g.,* +3.6 on DocVQA and +1.6 on InfoVQA. These results further validate the effectiveness of our mixture-of-resolution design on text-rich image understanding tasks.

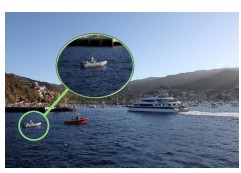

**User:** How many boats are fully visible?
**LLaVA-HR** (384 pix): 2
**LLaVA-HR** (1024 pix): 3

**User:** What color is the tiniest boat in the forefront?
**LLaVA-HR** (384 pix): Red
**LLaVA-HR** (1024 pix): White

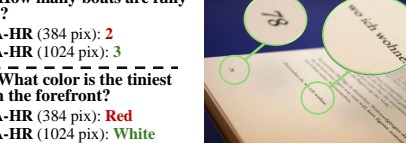

**User:** what are the last three words on this page?
**LLaVA-HR** (384 pix): "and other stories."
**LLaVA-HR** (1024 pix): Wo ich wohne.

**User:** what page is this book on?
**LLaVA-HR** (384 pix): This is the first page of a book.
**LLaVA-HR** (1024 pix): This book is on page 87.

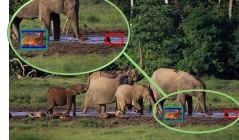

**User:** Please provide the bounding box coordinate of the region this sentence describes: The deer on the far right.
**LLaVA-HR** (384 pix): [0.92, 0.69, 0.97, 0.72]
**LLaVA-HR** (1024 pix): [0.66, 0.68, 0.74, 0.73]
Ground-truth: [0.66, 0.68, 0.74, 0.74]

**User:** Please provide the bounding box coordinate of the region this sentence describes: The child sliding down in red clothes.
**LLaVA-HR** (384 pix): [0.4, 0.6, 0.51, 0.78]
**LLaVA-HR** (1024 pix): [0.53, 0.57, 0.6, 0.71]
Ground-truth: [0.52, 0.56, 0.6, 0.7]

**(a) Comparison of LLaVA-HR with different resolutions**

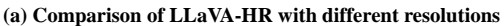

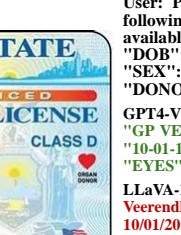

**User:** Please read the text in this image and return the information in the following JSON format (note xxx is placeholder, if the information is not available in the image, put "N/A" instead). {"class": XXX, "DLN": Xxx, "DOB": Xxx, "Name": XXX, "Address": XXX, "EXP": XXX, "ISS": XXX, "SEX": XXX, "HGT": XXX, WGT: XXX, "EYES": XXX, "HAIR": XXX, "DONOR": XXX}

**GPT4-V:** {"class": "D", "DLN": "97542954****", "DOB": "15-03-1984", "Name": "GP VEERE*****", "Address": "52 KLINK RD ROCHESTER ** *****", "EXP": "10-01-16", "ISS": "09-30-08", "SEX": "M", "HGT": "5-09", "WGT": "N/A", "EYES": "BR", "HAIR": "N/A", "DONOR": "YES"}

**LLaVA-1.5:** {"class": Class D, "DLN": 97745434, "DOB": 03/15/1988, "Name": Gopal Veerendhar, "Address": 54 Klinck Rd, Rochester, NY 14607, "EXP": 10/01/2016 - 10/01/2017, "ISS": New York State Department of Motor Vehicles, "SEX": Male, "HGT": 5'10", "WGT": 170, "EYES": Brown, "HAIR": Black, "DONOR": N/A}

**LLaVA-HR-X:** {"class": Class D, "DLN": 97542954****, "DOB": 15/03/1984, "Name": GP Veere*****, "Address": 52 Klink Rd, Rochester ** *****, "EXP": 10/01/2016, "ISS": New York State, "SEX": Male, "HGT": N/A, "WGT": N/A, "EYES": Brown, "HAIR": Black, "DONOR": N/A}

**(b) Comparison of LLaVA-HR-X, LLaVA-1.5 and GPT4-V in visual information extraction**

Figure 4: **Visualizations of LLaVA-HR and existing MLLMs.** Subfig-(a) shows that high image resolution greatly improves the capability of MLLMs on fine-grained VL tasks. In Subfig-(b), LLaVA-HR-X demonstrates the comparable ability with GPT4-V in visual information extraction. Correct and incorrect answers are colored in green and red, respectively.

### 5.3.2 QUALITATIVE EXPERIMENTS

In Fig 4 (a), we compare the predictions of LLaVA-HR with different resolutions. The visualizations show that higher image resolution obviously improves the capability of MLLMs on fine-grained tasks. For example, LLaVA-HR with a resolution of 1,024 × 1,024 can well capture granular visual content, *e.g.,* the tiny boat in the first example. Besides, high image resolution also enables LLaVA-HR a stronger ability of text recognition. For instance, the small and blurred phrase of "*wo ich wohne*" in the second example are correctly identified by the high-resolution LLaVA-HR. These results greatly confirm the significance of high image resolution in addressing visual shortcoming. In Fig 4 (b), we further compare the predictions of LLaVA-HR-X, LLaVA-1.5 (Liu et al., 2023a) and GPT4-V (OpenAI, 2023) in visual information extraction. Notably, LLaVA-HR-X shows a comparable ability with GPT4-V on this challenging task. As shown in Fig 4 (b), LLaVA-HR-X and GPT4-V can correctly extract almost all visual content of the driver license and organize it in JSON format. Compared to GPT4-V, LLaVA-HR-X also correctly identifies the hair color of the person, which requires fine-grained visual reasoning. In contrast, LLaVA-1.5 can only recognize simple visual content like "*class*" and "*SEX*", and fail to extract most visual information. These results further validate the effectiveness of MRA in addressing visual shortcoming of MLLMs.

## 6 CONCLUSION

In this paper, we focus on the efficient high-resolution adaptation for MLLMs and propose a novel method, namely *mixture-of-resolution adaptation* (MRA). MRA adopts dual visual pathways to process images of both high and low resolutions, where high-resolution information is embedded into the low-resolution modeling via the novel *mixture-of-resolution adapters* (MR-Adapters). We apply MRA to a popular MLLM called LLaVA-1.5, and construct a new high-resolution MLLM, termed LLaVA-HR. Experimental results not only validate the effectiveness of LLaVA-HR in addressing visual shortcoming, but also confirm its remarkable efficiency against existing MLLMs.

**Acknowledgments.** This work was supported by the National Science Fund for Distinguished Young Scholars (No.62025603), the China Postdoctoral Science Foundation (No. 2024M761548), the National Natural Science Foundation of China (No. U21B2037, No. U22B2051, No. 623B2088, No. U23A20383, No. U21A20472, No. 62176222, No. 62176223, No. 62176226, No. 62072386, No. 62072387, No. 62072389, No. 62002305 and No. 62272401), the Natural Science Foundation of Fujian Province of China (No. 2021J06003, No.2022J06001) and the Fundamental Research Funds for the Central Universities (Xiamen University: No. 20720240053).

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
