# Feast Your Eyes: Mixture-of-Resolution Adaptation for Multimodal Large Language Models

## Supplementary Material

## A    MORE EXPERIMENTS

**More ablation studies of MR-Adapter.** In Tab. 1, we conduct more ablations for MR-Adapter. From this table, the first observation is that our MRA performs better than the common choice, *i.e.,* cross-attention layer. In terms of other designs like fusion type, adapter structure, and gate function, we find that these micro-designs impact less on the final performance, and we adopt the best choice from these empirical studies. Overall, these results further validate the micro-designs of MR-Adapter.

**Results of combing MRA and DyRes.** To further improve the resolution, LLaVA-HR can be seamlessly combined with the dynamic high-resolution (DyRes) strategy of LLaVA-NeXT. As shown in Tab. 2, when combined with DyRes, LLaVA-HR can further boost its performance on the OCR-related benchmark, *i.e.,* +3.8% on TextVQA. These results confirm the generalization ability of MRA on larger resolutions.

**More results on OCR-related benchmarks.** To better understand the gains of LLaVA-HR on OCR-related benchmarks, we conduct fair comparisons with LLaVA-1.5 in Tab. 3. From these results, we can see that on five datasets, LLaVA-HR demonstrates obvious gains against LLaVA-1.5, *e.g.,* +8.9 on TVQA and +27.1 on DocVQA. These results not only confirm the significance of high resolutions, but also validate the design of LLaVA-HR on these benchmarks.

Table 1: **More ablation studies of MR-Adapter.** "Cross Attn." denotes the cross-attention layer. Our final setting is colored in gray.

| Settings | Choices | VQAv2 | TextVQA | MME | POPE |
|---|---|---|---|---|---|
| Adapter Choice | MRA | 81.8 | 64.4 | 1524.8 | 88.0 |
| | Cross Attn. | 80.9 | 64.0 | 1548.8 | 87.5 |
| Fusion Type | Sum | 81.8 | 64.4 | 1524.8 | 88.0 |
| | Concat | 81.7 | 64.7 | 1508.8 | 87.3 |
| Adapter Structure | mlp-conv | 81.8 | 64.4 | 1524.8 | 88.0 |
| | conv-conv | 81.6 | 64.6 | 1499.0 | 87.7 |
| | conv-mlp | 81.5 | 64.2 | 1517.9 | 87.6 |
| Gate Function | Tanh | 81.8 | 64.4 | 1524.8 | 88.0 |
| | Sigmoid | 81.7 | 64.3 | 1567.9 | 86.9 |
| | H-sigmoid | 81.6 | 64.4 | 1525.9 | 87.8 |

Table 2: **Comparison of LLaVA-HR and LLaVA-1.5 on OCR-related benchmarks.**

| Models | Param | TVQA | DocVQA | InfoVQA | AI2D | ChartQA |
|---|---|---|---|---|---|---|
| LLaVA-1.5 | 7B | 58.2 | 28.1 | 25.6 | 55.2 | 18.2 |
| LLaVA-HR | 7B | 67.1 | 45.2 | 29.3 | 55.8 | 24.0 |
| LLaVA-1.5 | 13B | 61.3 | 30.2 | 29.3 | 59.2 | 18.2 |
| LLaVA-HR | 14B | 70.9 | 52.2 | 34.5 | 59.7 | 27.6 |

Table 3: **Results of combing MRA and DyRes.**

| Models | Method | Res | VQAv2 | TVQA | MME | PoPE |
|---|---|---|---|---|---|---|
| LLaVA-HR | MRA | 1024 | 81.9 | 67.1 | 1554 | 87.6 |
| LLaVA-NeXt | DyRes | 1344 | 81.8 | 64.9 | 1519 | 86.5 |
| LLaVA-HR-NX | MRA+DyRes | 3072 | 81.9 | 70.9 | 1450 | 88.0 |

## B  MORE VISUALIZATIONS

"Please read the text in this image and return the information in the following JSON format (note xxx is placeholder, if the information is not available in the image, put N/A instead). {class: XXX，ID: Xxx，ISSUED: Xxx，Name: XXX，Address: XXX，EXP: XXX，ISS: XXX，SEX: XXX，HGT: XXX, WGT: XXX，EYES: XXX，HAIR: XXX，DONOR: XXX}"

**LLaVA-v1.5-7b**: Answer: {class: "Class D", ID: "Richard Fisher", ISSUED: "01/01/2003", Name: "Richard Fisher", Address: "Rest Street", EXP: "09/30/2011", DOB : "New York State", SEX: "M", HT: "5'10", EYES: "Brown", DONOR: "Richard Fisher"}

**+MRA**: Answer: {"class": "Class D","ID": " Richard Fisher ","ISSUED": " 01/10/2003","Name": "Richard Fisher","Address": "7384 Forest Street. Bronx, NY 10472.","EXP": "10/10/16","DOB": " New York State ","SEX": "M","EYES": "Brown","HT": "5'0-9'","DONOR": " Richard Fisher "}

**+1024 resolution**: Answer: {class: Class D, ID: 01234678, ISSUED: 09-03-08, Name: Richard Fisher, Address: 7384 Forest Street, Bronx, NY 10472, EXP: 10-16-16, DOB: 05-09-59, SEX: M, HT: 5'9",EYES: Brown, DONOR: None}

**+13B LLM**: Answer: {"class": "D","ID": "0123 456 789","ISSUED": "09/30/2008","Name": "Richard Fisher","Address": "7384 Forest Street. Bronx, NY 10472.","EXP": "10/10/2016","DOB": "05/09/1970","SEX": "M","EYES": "Brown","HT": "5'0-9'","DONOR": "Organ Donor"}

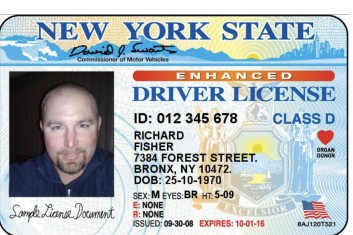

"Using the image showing a series of arithmetic problems, follow these instructions: Multiply the results of the first (top left) and last equations(bottom right)."

**LLaVA-v1.5-7b:** To multiply the results of the first (top left) and last equations (bottom right), you need to find the numbers that correspond to the same position in each equation. In this case, the numbers are 4 and 9. To multiply these numbers, you would write: 4 \* 9 = 36. So, the result of multiplying the first and last equations is 36.

**+MRA:** To multiply the results of the first equation (top left) and the last equation (bottom right), we need to find the products of the numbers in each equation. The first equation is: 4 + 7 = 11。 The last equation is: 3 × 8 = 24. Multiplying the results, we get: 11 × 24 = 264. So, the product of the first and last equations is 264.

**+1024 resolution**: To multiply the results of the first (top left) and last equations (bottom right), we need to find the common factors between the numbers. In this case, the common factors are 2 and 3. We can then multiply the results by these common factors. For the top left equation, 4 + 7 = 11, we multiply 11 by 2 to get 22. For the bottom right equation, 3 + 8 = 11, we multiply 11 by 3 to get 33. So, the result of multiplying the first and last equations is 22 + 33 = 55.

**+13B LLM:** To multiply the results of the first and last equations, we need to find the product of the numbers in the top left equation (4 + 7) and the bottom right equation (3 + 8). The top left equation is 4 + 7 = 11. The bottom right equation is 3 + 8 = 11. Since both equations result in the same value (11), we can simply multiply the numbers in the two equations:11 x 11 = 121. So, the result of multiplying the first and last equations is 121.

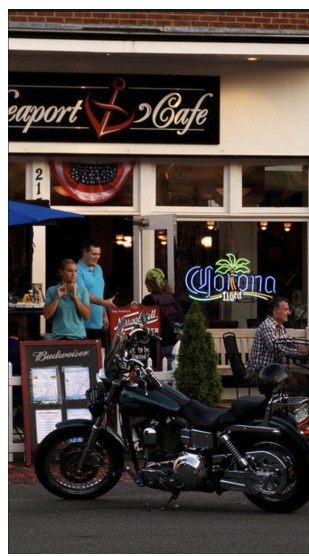

Question:
Detect all the three brands(the texts) in the image:
1. 2. 3.

**LLaVA-v1.5-7b:** 1. Seaport Cafe 2. Cuervo 3. Harley Davidson
**+MRA:** 1. Corona, 2. Harley Davidson, 3. Seaport Cafe
**+1024 resolution**: 1. Seaport Cafe, 2. Budweiser, 3. Yellowtail
**+13B LLM:** 1. Corona Light, 2. Budweiser, 3. Seaport Cafe

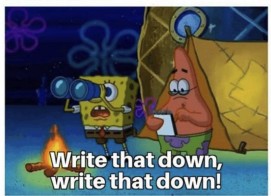

Question: "What are the characters and which tools are they using in the image?"

**LLaVA-v1.5-7b:** In the image, there are two characters: a professor and a freshman. The professor is using a telescope, while the freshman is using a microscope.
**+MRA:** The characters in the image are a yellow cartoon character (possibly a yellow sponge) and a red cartoon character (possibly a red crab or Squidward). They are using a magnifying glass and a clipboard as tools in the image.
**+1024 resolution**: In the image, there are two characters: a cartoon fish and a cartoon squid. The fish is holding binoculars, while the squid is holding a notebook.
**+13B LLM**: In the image, there are two cartoon characters: Spongebob and Patrick. Spongebob is holding binoculars, while Patrick is holding a clipboard.