# OpenReview forum: "Feast Your Eyes:  Mixture-of-Resolution Adaptation for Multimodal Large Language Models"
_ICLR.cc/2025/Conference — ICLR 2025 Poster_

### Official Review · Reviewer_Yq9F · 2024-10-21

**Soundness:** 3
**Presentation:** 3
**Contribution:** 3
**Rating:** 6
**Confidence:** 4

**Summary:**

This paper introduces an novel high-resolution adaptation method for multimodal large language models (MLLMs), termed Mixture-of-Resolution Adaptation (MRA). MRA employs a dual visual pathway design to process high- and low-resolution images simultaneously from both macro and micro perspectives, while integrating high-resolution information into the low-resolution pathway through the Mixture-of-Resolution Adapter (MR-Adapter). This approach reduces the number of visual tokens while preserving rich visual semantics, significantly enhancing the model's visual descriptive power.

**Strengths:**

- Unlike previous strategies that divide high-resolution images into sub-images, this paper introduces an innovative dual visual pathway structure, offering a fresh perspective for high-resolution adaptation. The MR-Adapter effectively embeds high-resolution information into the low-resolution pathway, introducing a new adaptation mechanism within the visual processing framework of MLLMs. This design overcomes the efficiency limitations of traditional high-resolution processing.

- The paper conducts extensive experiments across multiple vision-language tasks, providing a range of comparisons, with promising results.

- The writing is clear and easy to follow. It effectively highlights MRA's performance gains and efficiency advantages across different tasks, helping readers fully understand the model’s effectiveness and strengths.

**Weaknesses:**

1. The processing of both low-resolution and high-resolution images in the paper is mainly square-based, such as 448x448 and 1024x1024. Is there any adaptation mechanism for handling images with different aspect ratios? Would processing high-resolution images in a way that matches the input image's aspect ratio lead to better performance?

2. For high-resolution image inputs, we are more focused on improvements in OCR-related tasks. The results for OCRVQA in Table 5 don’t seem to be the best. Additionally, Table 6 only presents results for LLaVA-HR+, but it lacks results for LLaVA-HR-7B, LLaVA-HR-13B, and LLaVA-HR-X with less training data. It would be helpful to include these results to better illustrate the impact of MRA on OCR-related tasks.

3. Could the authors further explain why the MR-Adapter is inserted in the last 3 stages? What is the design principle behind this decision? Could it be inserted in the earlier stages instead?

**Questions:**

See Weakness.

---

> ### Author Response · Authors · 2024-11-22
>
> ------
>
> >**Comment#1** ： **The processing of both low-resolution and high-resolution images in the paper is mainly square-based, such as 448x448 and 1024x1024. Is there any adaptation mechanism for handling images with different aspect ratios? Would processing high-resolution images in a way that matches the input image's aspect ratio lead to better performance?**
>
>
>
> **Response**:  We appreciate for this professional comment.  In practice, we have already preserved the aspect ratio of the image and padded the short sides with zeros.  Nevertheless, your advice also inspires us to combine MRA with existing dynamic high-resolution methods [A].  By doing so, MRA not only achieves adaptation to arbitrary aspect ratios, but also further increases the resolution to 3k. As you expected, the model performance is further improved on TVQA and PoPE.
>
> | Model           | Res  | VQAv2 | TVQA | MME  | PoPE |
> | --------------- | ---- | ----- | ---- | ---- | ---- |
> | LLaVA-HR        | 1024 | 81.9  | 67.1 | 1554 | 87.6 |
> | LLaVA-HR+DyRes  | 3072 | 81.9  | 70.9 | 1450 | 88.0 |
>
> [A] Haotian Liu, Chunyuan Li, Yuheng Li, Bo Li, Yuanhan Zhang, Sheng Shen, and Yong Jae Lee. Llava-next: Improved reasoning, ocr, and world knowledge, January 2024a
>
>
>
> ------
>
> >**Comment#2 ：For high-resolution image inputs, we are more focused on improvements in OCR-related tasks. The results for OCRVQA in Table 5 don’t seem to be the best. Additionally, Table 6 only presents results for LLaVA-HR+, but it lacks results for LLaVA-HR-7B, LLaVA-HR-13B, and LLaVA-HR-X with less training data. It would be helpful to include these results to better illustrate the impact of MRA on OCR-related tasks.**
>
>
>
> **Response**: Thanks for your careful review.  In Table 5, Qwen-VL uses much more OCR-related data than LLaVA-HR, so it performs slightly better on OCRVQA, i.e., +1.5%.  For this reason, we make a more fair comparison in Tab 6, where LLaVA-HR uses similar or fewer data than existing methods.  In Tab 6,    even with less model size and training data,  LLaVA-HR still outperforms existing  methods like DocOwl-1.5-Chat on all OCR-related tasks.
>
> Moreover, we also fully agree that the LLaVA-HR with less data should also be compared with existing methods on these OCR-related data.  Thus, we provide the apple-to-apple comparison between LLaVA-1.5 and LLaVA-HR in the table below, where the training data  and the LLM are kept the same.  From this table, the benefit of MRA can still be observed on all OCR-related tasks.  We will update these results in our final revision.
>
>
>
> | Model              |   TVQA   |  DocVQA  | InfoVQA  |   AI2D   | ChartQA  |
> | ------------------ | :------: | :------: | :------: | :------: | :------: |
> | LLaVA-1.5-7B       |   58.2   |   28.1   |   25.6   |   55.2   |   18.2   |
> | **LLaVA-HR-7B**    | **67.1** | **45.2** | **29.3** | **55.8** | **24.0** |
> | LLaVA-1.5-13B      |   61.3   |   30.2   |   29.3   |   59.2   |   18.2   |
> | **LLaVA-HR-X-14B** | **70.9** | **52.5** | **34.5** | **59.7** | **27.6** |
>
> ------
>
> >**Comment#3：Could the authors further explain why the MR-Adapter is inserted in the last 3 stages? What is the design principle behind this decision? Could it be inserted in the earlier stages instead?**
>
>
>
> **Response**:  Great question!  We think that MR-Adapter should not be inserted in earlier stages for two reasons:
>
> 1. The early stages of ViT usually aim to encode low-level visual information, which is inefficient for grasping high-level semantic and fine-grained information from the features of ConvneXt.
> 2. Since the early stage of ViT has not yet extracted high-level semantics, the early fusion of ConvneXt may hurt the original feature semantics of ViT.
>
> In Tab 3, we have already conducted detailed ablations to validate the insert position of MR-Adapter, which also confirms that the last 3 stages is the optimal choice.
>
> | Insert Pos         | VQAv2    | TVQA     | MME      | PoPE     |
> | ------------------ | -------- | -------- | -------- | -------- |
> | **last  3 stages** | **81.8** | **64.4** | **1524** | **88.0** |
> | last stage         | 81.3     | 62.8     | 1513     | 87.2     |
> | last 2 stages      | 81.6     | 63.8     | 1508     | 87.5     |
> | last 4 stages      | 81.4     | 63.1     | 1461     | 87.5     |
>
> ------

---

> > ### Author Response · Authors · 2024-11-25
> >
> > Dear reviewer Yq9F,
> >
> > Thanks again for your valuable time and insightful comments. As the deadline for the Author/Reviewer discussion is approaching, it would be nice of you to let us know whether our answers have solved your concerns so that we can better improve our work. We are happy to provide any additional clarifications that you may need.
> >
> > Best regards!

---

> > > ### Comment · Reviewer_Yq9F · 2024-11-25
> > >
> > > The authors have addressed my concerns, and I maintain my positive score of this paper.

---

> > > > ### Author Response · Authors · 2024-11-25
> > > >
> > > > Thanks for your encouraging comment. We would like to appreciate again for your valuable suggestions.

---

### Official Review · Reviewer_HwDF · 2024-10-28

**Soundness:** 3
**Presentation:** 3
**Contribution:** 3
**Rating:** 6
**Confidence:** 4

**Summary:**

This paper focuses on the efficient high-resolution adaptation for multimodal large language models (MLLMs) and proposes a mixture-of-resolution adaptation (MRA) method for MLLMs. To be specific, the proposed MRA employs two visual pathways for images of different resolutions, where high-resolution visual information is embedded into the low-resolution pathway via the mixture-of-resolution adapters. Besides, the paper conducts extensive experiments to verify the effectiveness of the proposed model.

**Strengths:**

1. The paper aims to explore the high-resolution adaptation for MLLMs, which is crucial and engaging.
2. The paper is well written and easy to follow.
3. The paper is well motivated and the proposed MRA appears reasonable.

**Weaknesses:**

1. As demonstrated in Table 1, it seems that there is no significant gap between ‘Avg. Pooling’ and the proposed MRA for the VQAv2 task, which is perplexing. The paper should explain the experimental phenomenon.
2. The paper should carry out a qualitative experiment between the proposed MRA and the model variant in Table 2.
3. The paper fails to clarify the version of LLaVA-1.5 used in Figure 4.

**Questions:**

As mentioned, in Table 1, it seems that there is no significant gap between ‘Avg. Pooling’ and the proposed MRA for the VQAv2 task, which is perplexing. The paper should explain the experimental phenomenon.
2. The paper should carry out a qualitative experiment between the proposed MRA and the model variant in Table 2.

---

> ### Author Response · Authors · 2024-11-22
>
> ------
>
> >**Comment#1 ：As demonstrated in Table 1, it seems that there is no significant gap between ‘Avg. Pooling’ and the proposed MRA for the VQAv2 task, which is perplexing. The paper should explain the experimental phenomenon.**
>
>
>
> **Response**: Thanks for this comment. We would like to explain this from two aspects:
>
> 1.  **In a fair comparison setting, performance gains of MRA are  indeed noticeable on VQAv2, i.e., +1.3% over  ‘Avg. Pooling’.**  In practice, improving VQAv2 performance to above 80 is quit challenging. To the best of our knowledge, PaLI-X (55B) achieves state-of-the-art VQA performance of 86.0, which is only 3.7% higher than our much smaller model (LLaVA-HR-X, 14B).
> 2. **Most images of VQAv2 are low-  and middle-resolution ones, so  the high-resolution benefit of MRA cannot be fully reflected in VQAv2.**   This is also evidenced by the larger gains of MRA in more fine-grained benchmarks such as 7.5% on TextVQA.
>
> Following your suggestion, we will add these discussions in our final version.
>
>
>
> ------
>
> >**Comment#2** ： **The paper should carry out a qualitative experiment between the proposed MRA and the model variant in Table 2.**
>
>
>
> **Response**:  Thanks for this constructive suggestion. We fully agree your advice and have provided several visualization examples in our appendix. We believe that these comparisons do contribute to the understanding of our paper.
>
>
>
> ------
>
> >**Comment#3： The paper fails to clarify the version of LLaVA-1.5 used in Figure 4.**
>
>
>
> **Response**:  Thank you for your careful review. We apologize for the missing model details in Figure 4. In fact, we use LLaVA-1.5-13B from the official checkpoint  for comparison, and its LLM is the same as LLaVA-HR-X.  According to your advice, we will add more details in our final version.
>
> ------

---

> > ### Author Response · Authors · 2024-11-25
> >
> > Dear reviewer HwDF,
> >
> > Thanks again for your valuable time and insightful comments. As the deadline for the Author/Reviewer discussion is approaching, it would be nice of you to let us know whether our answers have solved your concerns so that we can better improve our work. We are happy to provide any additional clarifications that you may need.
> >
> > Best regards!

---

> > > ### Comment · Reviewer_HwDF · 2024-11-26
> > >
> > > I am sorry for the late response. The authors have addressed my concerns, and I maintain my positive score of this paper.

---

> > > > ### Author Response · Authors · 2024-11-26
> > > >
> > > > Thank you for your support! We are pleased to address your concerns and greatly appreciate your effort in helping us to strengthen our work.
> > > >
> > > > Best regards!

---

### Official Review · Reviewer_Mfi4 · 2024-10-28

**Soundness:** 3
**Presentation:** 3
**Contribution:** 2
**Rating:** 5
**Confidence:** 4

**Summary:**

This paper presents a new approach for efficient multimodal large language models (MLLMs) by addressing the high computational cost of processing high-resolution images. The authors introduce Mixture-of-Resolution Adaptation (MRA), a method that combines both low- and high-resolution visual features to enhance model efficiency without compromising visual recognition quality. MRA uses two visual pathways: one for low-resolution and one for high-resolution images, with novel mixture-of-resolution adapters (MR-Adapters) that embed high-resolution information into the low-resolution pathway. This design significantly reduces input sequence length and computational load.

The authors apply MRA to the LLaVA model, resulting in an improved version called LLaVA-HR, which demonstrates superior performance across 15 out of 17 vision-language (VL) tasks, including a 5.2% increase in accuracy on TextVQA. Furthermore, LLaVA-HR maintains efficient training and inference times, showing improvements over LLaVA-NeXT.

**Strengths:**

1. The paper is well-written and easy to follow.

2. Figures 2 and 3 are effectively designed and enhance understanding of the framework.

3. The ablation study is solid to reveal the contribution of component.

**Weaknesses:**

> ### 1. LImited performance imprvement.

The performance gains with MRA are modest. The low-resolution branch operates at 448×448, so the appropriate baseline is LLaVA-1.5 with 448-pixel resizing. Compared to this baseline, the improvements MRA achieves are minimal (e.g., +0.7 on VQA v2, +31 on MME, and +0.8 on POPE). Training cost and inference speed are also similar between MRA and LLaVA-1.5-448, reducing the practical benefit.

> ### 2. Limited novelty

The dual-pathway, high-and-low-resolution approach isn’t particularly new. Similar strategies have been explored in other works, such as Mini-Gemini and CogAgent, yet the authors do not compare their method with these models. Explicitly differentiating MRA from these approaches would help clarify its unique contributions.

> ### 3. Limited generalizability

The authors apply MRA solely to LLaVA-1.5. Expanding the evaluation to other MLLMs, like Qwen-VL, would strengthen claims of the method’s generalizability across architectures.


[1] CogAgent: A Visual Language Model for GUI Agents
[2] Mini-Gemini: Mining the Potential of Multi-modality Vision Language Models

**Questions:**

> ### 1. Clarification on Visual Encoder Notation

In line 206, it states that $F_{I_l}$ and $F_{I_h}$ are visual encoders for high- and low-resolution images, which seems to be a typo. The correct notation should reflect that $F_{I_l}$ and $F_{I_h}$ correspond specifically to low- and high-resolution encoders, respectively.

> ### 2. MR-Adapter Placement in ViT Architecture

Figure 2 shows the MR-Adapter is applied starting from the second stage of the ViT architecture. Does this mean the initial stage of the ViT does not utilize high-resolution features? Clarifying this could help illustrate the feature extraction flow more clearly.

> ### 3. Implementation of LLaVA-1.5-448

For LLaVA-1.5-448, only the image resolution is modified at the fine-tuning stage. Have you considered modifying the visual backbone from ViT-336 to ViT-448 and retraining it for both pre-training and fine-tuning? This comparison could provide insight into performance differences when using higher resolution throughout the model’s entire training process.

> ### 4. $SEED^{img}$ Performance Comparison

Could you provide the $SEED^{img}$ performance for LLaVA-1.5, LLaVA-1.5-448, and LLaVA-NeXT? This metric would help evaluate relative image-processing capabilities across these models.

---

> ### Author Response · Authors · 2024-11-22
>
> ------
>
> >**Comment#1: LImited performance imprvement: The performance gains with MRA are modest. The low-resolution branch operates at 448×448, so the appropriate baseline is LLaVA-1.5 with 448-pixel resizing. Compared to this baseline, the improvements MRA achieves are minimal (e.g., +0.7 on VQA v2, +31 on MME, and +0.8 on POPE). Training cost and inference speed are also similar between MRA and LLaVA-1.5-448, reducing the practical benefit.**
>
>
>
> **Response**:  Thanks for this comment. We fully agree that LLaVA-1.5-448 should be a suitable baseline for LLaVA-HR-1024. As you said,  with similar costs, LLaVA-HR can already achieve varying degrees of gain on low-resolution or medium-resolution benchmarks such as VQAv2 and MME. But as a high-resolution method, more gains of LLaVA-HR should be observed on high-resolution benchmarks such as TextVQA and DocVQA. To help you better understand our contribution, we provide an apple-to-apple comparison on these benchmarks in the table below, which shows the clear gains of LLaVA-HR-1024 over LLaVA-1.5-448.
>
> | Model            | TVQA     | DocVQA   | InfoVQA  | AI2D     | ChartQA  |
> | ---------------- | -------- | -------- | -------- | -------- | -------- |
> | LLaVA-1.5-7B-448 | 62.1     | 30.3     | 26.8     | 55.1     | 18.4     |
> | LLaVA-HR-7B-1024 | **67.1** | **45.2** | **29.3** | **55.8** | **24.0** |
>
> ------
>
> >**Comment#2: Limited novelty: the dual-pathway, high-and-low-resolution approach isn’t particularly new. Similar strategies have been explored in other works, such as Mini-Gemini and CogAgent, yet the authors do not compare their method with these models. Explicitly differentiating MRA from these approaches would help clarify its unique contributions.**
>
>
>
>
> **Response**:  Thanks for this kindly suggestion. We would like to recognize that  Mini-Gemini, as the concurrent  work to LLaVA-HR, do have the similar idea in dual visual pathways.  However,  in terms of micro designs, LLaVA-HR is quit different and efficient against Mini-Gemini.  From the comparison in Tab 4, we can see that LLaVA-HR with 1,024 visual tokens can outperform MiniGemini with 2880 visual tokens on 5 of 6 benchmarks.
>
>
>
> Compared to CogAgent,  LLaVA-HR still establishes advantage  in simplicity and efficiency.  For example, the high-resolution cross-module of CogAgent requires a large amount of data for pre-training, while our MRA does not.   To further validate the benefit of LLaVA-HR, we would like to provide a relatively fair comparison on high-resolution benchmarks in the table below, where CogAgent uses much more training data.  From this table, we also see the better performance of LLaVA-HR than CogAgent on 3 of 4 benchmarks.
>
>
>
> Your advice is highly beneficial to our paper, and all comparisons will be added in our final version.
>
>
>
> | Model                     | TVQA | DocVQA | InfoVQA | ChartQA |
> | ------------------------- | ---- | ------ | ------- | ------- |
> | CogAgent                  | 76.1 | 81.6   | 44.5    | 68.4    |
> | LLaVA-HR-7B-1024$\dagger$ | 73.8 | 85.8   | 52.3    | 77.6    |
>
> ------
>
>
> >**Comment#3:  Limited generalizability: the authors apply MRA solely to LLaVA-1.5. Expanding the evaluation to other MLLMs, like Qwen-VL, would strengthen claims of the method’s generalizability across architectures.**
>
>
>
>
>
> **Response**:  We fully respect to your concerns regarding the generalizability.  However, LLaVA-based architecture has almost become the mainstream paradigm of existing MLLMs, thus LLaVA-1.5 may be the most representative and generalizable baseline.  Based on your concerns,  we have tried to combine MRA with LLaVA-NeXT, another reprehensive MLLM Architecture with dynamic high-resolution strategy.  By adopting MRA  to  each dynamic patch for feature extraction, we observe additional gains on TVQA and PoPE.
>
> | Model           | Res  | VQAv2 | TVQA | MME  | PoPE |
> | --------------- | ---- | ----- | ---- | ---- | ---- |
> | LLaVA-HR        | 1024 | 81.9  | 67.1 | 1554 | 87.6 |
> | LLaVA-NeXT      | 1344 | 81.8  | 64.9 | 1519 | 86.5 |
> | LLaVA-NeXT+MRA  | 3072 | 81.9  | 70.9 | 1450 | 88.0 |
>
> ------
>
> >**Comment#4: Clarification on Visual Encoder Notation:  In line 206, it states that $\mathcal{F_{I}}l$ and $\mathcal{F_{I}}h$ are visual encoders for high- and low-resolution images, which seems to be a typo. The correct notation should reflect that $\mathcal{F_{I}}h$ and $\mathcal{F_{I}}l$ correspond specifically to low- and high-resolution encoders, respectively.**
>
>
>
> **Response**:  Thanks for your careful review  and  we will revise these typos in our final version. In addition to your mentioned typos, we will carefully revise our paper to improve the readability.
>
>
>
> ------

---

> > ### Author Response · Authors · 2024-11-22
> >
> > ---
> >
> >
> > >**Comment#5: MR-Adapter Placement in ViT Architecture:  figure 2 shows the MR-Adapter is applied starting from the second stage of the ViT architecture. Does this mean the initial stage of the ViT does not utilize high-resolution features? Clarifying this could help illustrate the feature extraction flow more clearly.**
> >
> >
> >
> > **Response**: Thanks for this professional comment.  We think that MR-Adapter should not be inserted in earlier stages for two reasons:
> >
> > 1. The early stages of ViT usually aim to encode low-level visual information, which is inefficient for grasping high-level semantic and fine-grained information from the features of ConvneXt.
> > 2. Since the early stage of ViT has not yet extracted high-level semantics, the early fusion of ConvneXt may hurt the original feature semantics of ViT.
> >
> > In Tab 3, we have already conducted detailed ablations to validate the insert position of MR-Adapter, which also confirms that the last 3 stages is the optimal choice.
> >
> > | Insert Pos     | VQAv2    | TVQA     | MME      | PoPE     |
> > | -------------- | -------- | -------- | -------- | -------- |
> > | last  3 stages | **81.8** | **64.4** | **1524** | **88.0** |
> > | last stage     | 81.3     | 62.8     | 1513     | 87.2     |
> > | last 2 stages  | 81.6     | 63.8     | 1508     | 87.5     |
> > | last 4 stages  | 81.4     | 63.1     | 1461     | 87.5     |
> >
> > ------
> >
> >
> >
> > >**Comment#6:  Implementation of LLaVA-1.5-448: For LLaVA-1.5-448, only the image resolution is modified at the fine-tuning stage. Have you considered modifying the visual backbone from ViT-336 to ViT-448 and retraining it for both pre-training and fine-tuning? This comparison could provide insight into performance differences when using higher resolution throughout the model’s entire training process.**
> >
> >
> >
> > **Response**:  We appreciate this constructive advice. In fact, jointly optimizing the visual encoder and randomly initialized MLP layers is technically tricky and requires carefully tuned learning rates and more training data.   In particular, we follow QwenVL to use a learning rate of 2e-4 to tune the ViT-448 in the pre-training stage, and observe that its performance is similar to the frozen one.  To this end, we think that the frozen visual encoder setting will be more simple and stronger in our experiments.
> >
> > | Model         | Stage-1 | VQAv2 | TVQA | MME  | PoPE |
> > | ------------- | ------- | ----- | ---- | ---- | ---- |
> > | LLaVA-1.5-448 | Fixed   | 80.4  | 59.4 | 1461 | 86.2 |
> > | LLaVA-1.5-448 | Tuned   | 80.4  | 59.2 | 1420 | 86.6 |
> > | LLaVA-HR-1024 | Fixed   | 81.8  | 64.4 | 1524 | 88.0 |
> >
> > ------
> >
> > >**Comment#7:  Seed$^I$ Performance Comparison: Could you provide the Seed$^I$ performance for LLaVA-1.5, LLaVA-1.5-448, and LLaVA-NeXT? This metric would help evaluate relative image-processing capabilities across these models.**
> >
> >
> >
> > **Response**:  Of course, we are glad to provide  the Seed$^I$ performance of LLaVA-1.5, LLaVA-1.5-448, LLaVA-NeXT and LLaVA-HR in the table below, which also confirms the superior performance of LLaVA-HR. We believe these results do benefit our paper and will be added them to the final version.
> >
> > | Split    | LLaVA-1.5 | LLaVA-1.5-448 | LLaVA-NeXT | LLaVA-HR-1024 |
> > | -------- | :-------: | :-----------: | :--------: | :-----------: |
> > | Seed     |   58.6    |     63.8      |     -      |     64.2      |
> > | Seed$^I$ |   66.1    |     69.8      |    70.2    |     70.6      |
> >
> >
> >
> > ------

---

> > > ### Author Response · Authors · 2024-11-23
> > >
> > > Dear reviewer Mfi4,
> > >
> > > Thanks again for your valuable time and insightful comments. As the deadline for the Author/Reviewer discussion is approaching, it would be nice of you to let us know whether our answers have solved your concerns so that we can better improve our work. We are happy to provide any additional clarifications that you may need.
> > >
> > > Best regards!

---

> > > > ### Author Response · Authors · 2024-11-25
> > > >
> > > > Dear reviewer Mfi4,
> > > >
> > > > We are sorry that this message may bother you again. We sincerely hope that you could take your valuable time to read our response. Since the discussion deadline is already approaching, we are worry that there will not be enough time to address your further concerns.
> > > >
> > > > Best regards!

---

> > > > > ### Author Response · Authors · 2024-11-28
> > > > >
> > > > > Dear reviewer Mfi4,
> > > > >
> > > > > We fully understand your busyness, and also sincerely hope that our efforts would be recognized by you. To date, we have received four positive scores by other reviewers, one of whom raised their scores to positive after reading our response. Thus,  we are still looking forward to your new decision based on our response. Thank you!
> > > > >
> > > > > Best regards!

---

> > > > > > ### Author Response · Authors · 2024-12-02
> > > > > >
> > > > > > Dear Reviewer Mfi4,
> > > > > >
> > > > > > We hope that our detailed rebuttal can address your concerns about this paper.
> > > > > >
> > > > > > As the deadline is approaching, we are looking forward to your valuable feedback and also welcome any new questions you may have.
> > > > > >
> > > > > > Thanks again for your time and efforts in reviewing this paper.
> > > > > >
> > > > > > Best regards,
> > > > > >
> > > > > > The authors

---

### Official Review · Reviewer_b6Tu · 2024-10-29

**Soundness:** 3
**Presentation:** 3
**Contribution:** 2
**Rating:** 6
**Confidence:** 4

**Summary:**

In this paper, the authors propose the Mixture-of-Resolution Adaptation method to embed the high-resolution features into the low-resolution pathway. The MRA enhances the visual perception ability in MLLMs, and allow them to benefit from high-resolution visual inputs with reduced computational cost. Extensive experiments demonstrate the effectiveness of the MRA.

**Strengths:**

1. The paper is well-written and easy to follow.
2. The comparison of MRA and other high-resolution adaptation solutions is clear, highlighting the effectiveness of the dual visual pathways.
3. The experiments are well-conducted and quite comprehensive.
4. The study demonstrates strong performance on most datasets compared with other MLLMs.

**Weaknesses:**

1. In Table 1, the MRA is compared to other high-resolution adaptation methods that use a single visual pathway. However, the introduction of a new visual encoder in the MRA raises concerns about the fairness of this comparison. Could the authors provide a baseline that uses dual visual pathways without the MR-Adapter?
2. The analyses of the MRA’s architecture and design details are insufficient, particularly regarding $\mathcal{F}_l$, $\mathcal{F}_h$, and the gate function. Could the authors provide ablation studies on these components?
3. The main novelty of the paper appears to be the Mixture-of-Resolution Adapter. While the application of dual visual pathways for high-resolution adaptation in MLLMs is innovative, the overall contribution of the paper seems somewhat insufficient. If MR-Adapter could integrate a wider variety of low- and high- resolution visual encoders, its contribution would be significantly enhanced.

**Questions:**

1. There are several micro-designs in the Mixture-of-Resolution Adapter, including $\mathcal{F}_l$, $\mathcal{F}_h$, and the gate function. Why do we choose a conv layer for $\mathcal{F}_l$, an MLP layer for $\mathcal{F}_h$? Are these layers and functions necessary? Please provide some analyses.

2. In the Mixture-of-Resolution Adapter, the authors choose the addition operation to fuse features of different resolutions. (Deformable) Cross Attention is also an option. I wonder which method is better?

---

> ### Author Response · Authors · 2024-11-22
>
> ------
> >**Comment#1**：  **In Table 1, the MRA is compared to other high-resolution adaptation methods that use a single visual pathway. However, the introduction of a new visual encoder in the MRA raises concerns about the fairness of this comparison. Could the authors provide a baseline that uses dual visual pathways without the MR-Adapter?**
>
>
>
> **Response:**  Thanks for this suggestion.  We fully respect your concerns and think that our dual-pathway designs including the MR-Adapter and the multi-resolution pathway (rather than the additional encoder) play  crucial roles in LLaVA-HR.    Therefore, we conduct  additional ablations in the table below, which shows that **the gain of the additional visual encoder is minor if our designs  are not used.**   We hope these comparisons can further eliminate your confusion.
>
> | Model     | VQAv2 | TVQA | MME  | PoPE |
> | --------- | ----- | ---- | ---- | ---- |
> | LLaVA-1.5 | 80.4  | 59.4 | 1461 | 86.2 |
> | +ConvNeXT | 80.4  | 59.6 | 1501 | 86.3 |
>
> ------
>
>
> >**Comment#2**： **The analyses of the MRA’s architecture and design details are insufficient, particularly regarding $\mathcal{F}_l$, $\mathcal{F}_h$ and the gate function. Could the authors provide ablation studies on these components?**
>
>
>
> **Response:**   Thanks for your detailed review. As discussed above,  our main focus and contribution are the macro designs of MRA  (the MR-Adapter and the multi-resolution pathway) ,  whose motivations and ablations are detailed in Sec 4.2-4.3 and Tab 1-2, respectively. As for the micro design of MRA, we aim to explore its optimal choice through empirical studies, and part results are already  listed in Tab 3  (including the impact of $\mathcal{F}_l$, $\mathcal{F}_h$, fusion direction and insert position).
>
>
>
> To further address your concerns, we provide more ablations of the gate function in the table below.   As shown in the table, **their significance is far from the macro design of MRA, so we may lack detailed discussions due to page limitations**. Based on your suggestion, we will add these results in our final version.
>
> | $\tau_h$     |   $\tau_l$     | VQAv2 | TVQA | MME  | PoPE |
> | --------- | ----- | ---- | ---- | ---- | ---- |
> | **mlp** | **conv** | **81.8** | **64.4** | **1524** | **88.0** |
> | conv | conv | 81.6  | 64.6 | 1499 | 87.7 |
> | conv | mlp | 81.5 | 64.2 | 1517 | 87.6 |
> | mlp | mlp | 81.5 | 64.1 | 1501 | 87.4 |
>
> | Gate Function | VQAv2 | TVQA | MME  | PoPE |
> | ----- | ---- | ---- | ---- | ---- |
> | **tanh** | **81.8** | **64.4** | **1524** | **88.0** |
> | sigmoid | 81.7  | 64.3 | 1567 | 86.9 |
> | H-sigmoid | 81.6 | 64.4 | 1525 | 87.8 |
>
> ------
>
>
> >**Comment#3**：**The main novelty of the paper appears to be the Mixture-of-Resolution Adapter. While the application of dual visual pathways for high-resolution adaptation in MLLMs is innovative, the overall contribution of the paper seems somewhat insufficient.**
>
>
>
> **Response:**  Thanks for this comment. To help you better understand our innovations, we would like to highlight the contribution of our dual-pathway design from two aspects.
>
> 1. **Design principle.**  As discussed in **Comment#1**, directly combing two visual pathways does not lead to obvious performance gains in MLLMs. Therefore, **the main advantage of the dual-pathway comes from our design principle, which  fully considers the visual complementarity of different encoders from the perspective of functionality  and alignmen**t, as described  in Sec 4.2.
> 2. **Technical details.** Previous works we compared in Table 4 (such as Sphinx) also mix multiple visual features, but their motivations and technical details are quit different. For example, Sphinx mixes visual embeddings for better representation, but still requires a dynamic high-resolution method for high-resolution encoding. **In contrast, we unify feature mixing and resolution enhancement into one dual-pathway design, greatly improving the efficiency.**
>
> Overall, we believe that the design principle and technical details of MRA will provide  good hints for future work.
>
> ------

---

> ### Author Response · Authors · 2024-11-22
>
> ---
> >**Comment#4：If MR-Adapter could integrate a wider variety of low- and high- resolution visual encoders, its contribution would be significantly enhanced.**
>
>
>
> **Response:**  Thanks for this advice.  We focus on high-resolution adaptation for MLLMs, thus two visual pathways can already efficiently achieve our target.  However, MR-Adapter can also be directly extended to more visual encoders. To validate this,  we conduct a toy experiment in the table below, which  further fuses features of SigLip into the CLIP-ViT ones.  From the table, experimental results also confirm the generalization ability of MR-Adapter.
>
> | Encoders | VQAv2 | TVQA | MME  | PoPE |
> | ----- | ---- | ---- | ---- | ---- |
> | CLIP + ConvNext | 81.8 | 64.4 | **1524** | **88.0** |
> | CLIP + ConvNext +SigLip | **82.0** | **65.5** | 1501 | 87.9 |
>
> ------
>
> >**Comment#5 ：In the Mixture-of-Resolution Adapter, the authors choose the addition operation to fuse features of different resolutions. (Deformable) Cross Attention is also an option. I wonder which method is better?**
>
>
>
> **Response:** Yes, cross attention is a viable choice for feature mixing. However, compared with MR-Adapter, cross attention requires longer training steps to converge and incurs more computational cost. Therefore, we adopt the simple yet effective design for MR-Adapter.
>
> | Fusion Module | VQAv2 | TVQA | MME  | PoPE |
> | ----- | ---- | ---- | ---- | ---- |
> | MR-Adapter | **81.8** | **64.4** | **1524** | **88.0** |
> | Cross Attention | 80.9 | 64.0 | 1483 | 87.5 |
>
> ------

---

> > ### Author Response · Authors · 2024-11-23
> >
> > Dear reviewer b6Tu,
> >
> > Thanks again for your valuable time and insightful comments. As the deadline for the Author/Reviewer discussion is approaching, it would be nice of you to let us know whether our answers have solved your concerns so that we can better improve our work. We are happy to provide any additional clarifications that you may need.
> >
> > Best regards!

---

> > > ### Author Response · Authors · 2024-11-25
> > >
> > > Dear reviewer b6Tu,
> > >
> > > We are sorry that this message may bother you again. We sincerely hope that you could take your valuable time to read our response. Since the discussion deadline is already approaching, we are worry that there will not be enough time to address your further concerns.
> > >
> > > Best regards!

---

> > > > ### Comment · Reviewer_b6Tu · 2024-11-26
> > > >
> > > > Thank you for the authors' response. Most of my concerns are addressed, and I have decided to increase my score.

---

> > > > > ### Author Response · Authors · 2024-11-26
> > > > >
> > > > > Thanks for your encouraging comment. We would like to appreciate again for your valuable suggestions, which play a crucial role in improving our work.
> > > > >
> > > > > Best regards!

---

### Official Review · Reviewer_TWQ4 · 2024-11-04

**Soundness:** 3
**Presentation:** 3
**Contribution:** 3
**Rating:** 6
**Confidence:** 4

**Summary:**

This paper aims to enhance MLLM by enlarging resolution of input images. By combining features from ViT and a CNN encoder through an adapter, performances of MLLM are improved a lot. Meanwhile, fusing high-resolution features from convolution-based encoder into low-resolution features from transformer-based encoder does not increase vision tokens to LLM decoder, so that additional computational cost is low. Proposed LLaVA-HR increases effective resolution for MLLM to 1024 and outperforms concurrent MLLMs.

**Strengths:**

This work proposed a novel method to increase resolutions of MLLMs, which is an important problem in the field and critical in fine-grained  vision tasks. Without large modification of training recipe and computational cost of its baseline, LLaVA-1.5.
Evalutions are conducted on many existing benchmarks and performance of LLaVA-HR is quite impressive. Besides, the computational cost involved is quite small compared with related works.

**Weaknesses:**

Please see as in questions.

**Questions:**

1. In section4.3(line 258), the statement, global average pooling is confusion, is the features are pooled into 1 global token? If so, it seems to be not consistent with figures. Please clarify the exact dimensions of fv after global average pooling.
2. In Table 1, resizing LLaVA-1.5 to 672 pix achieves close performance with 768pix version of LLaVA-HR, is there a direct comparision between 768-pix version of them?
3. In table 2, there is an ablation of "tune vision" referring to finetune vision encoder. However, I think the vision encoder in LLaVA-1.5 is fixed, can you provide a detailed description about this. For example, implementation and aim of tuning vision encoder.
4. LLaVA-HR is proposed to process input resolution of 1024, what if input images larger than 1024. Is there any extended experiments for even larger images such as 4K ones.
5. What do you mean by "stages" in vision transformers? And, currently only final features from ConvNext is utilized, is there any experiments of multi-stage feature integration for that of CNN encoder?

---

> ### Author Response · Authors · 2024-11-22
>
> ------
> >  **Comment#1： In section4.3 (line 258), the statement, global average pooling is confusion, is the features are pooled into 1 global token? If so, it seems to be not consistent with figures. Please clarify the exact dimensions of fv after global average pooling.**
>
>
>
> **Response:**  We feel sorry for any confusion of  Fig 3, which does not detail the pooling operation.  Actually, global average pooling is only used to compute the gating vector $g\in d$. In this process,  the concatenated visual features from dual pathways are  globally pooled to a vector $f_v \in \mathbb{R}^{2d}$, and produce the gating vector via Eq. 4.  We will add more details in Fig 3 to improve its readability.
>
> ------
>
>
>
> > **Comment#2： In Table 1, resizing LLaVA-1.5 to 672 pix achieves close performance with 768pix version of LLaVA-HR, is there a direct comparison between 768-pix version of them?**
>
>
>
> **Response:**   Yes, we are glad to provide the 756-pix version of LLaVA-1.5 for comparison. Note that 768 pixels does not evenly divide the stride of ViT ($14 \times 14$), so we choose 756 pixels for better performance.
>
> | Model        | Res      | V-token  | VQAv2    | TVQA     | MME      | Speed        |
> | ------------ | -------- | -------- | -------- | -------- | -------- | ------------ |
> | LLaVA-1.5    | 672      | 2304     | 81.5     | 64.2     | 1498     | 12.7 t/s     |
> | LLaVA-1.5    | 756      | 2916     | 81.0     | 63.2     | 1436     | 10.7 t/s     |
> | **LLaVA-HR** | **768**  | **576**  | **81.8** | **64.3** | **1524** | **23.5 t/s** |
> | LLaVA-1.5    | 1022     | 5329     | 74.2     | 37.8     | 1266     | 5.6 t/s      |
> | **LLaVA-HR** | **1024** | **1024** | **81.9** | **67.1** | **1554** | **19.7 t/s** |
>
> As shown in the table, a higher resolution of LLaVA-1.5 does not necessarily lead to higher performance, but it does incur a greater computational overhead.   In stark comparison, LLaVA-HR  achieves **2$\times$ faster inference speed** and  **higher performance**.  We hope that these results can better help you understand our contribution.
>
> ------
>
>
>
>
> > **Comment#3： In table 2, there is an ablation of "tune vision" referring to finetune vision encoder. However, I think the vision encoder in LLaVA-1.5 is fixed, can you provide a detailed description about this. For example, implementation and aim of tuning vision encoder.**
>
>
>
> **Response:**  Thanks for your detailed review. Actually, fine-tuning the vision encoder can usually improve performance, especially as image resolution increases.  **In this case, all of  baselines in ablations (including LLaVA-1.5) adopt "tune vision"  as the default setting, thus providing a fair and strong comparison with LLaVA-HR.**
>
>
>
> To further address your concerns, we provide the ablation of "tune vision" for LLaVA-1.5 in the table below, which shows that stronger baseline performance can be obtained via "tune vision".
>
> | Model         | Res     | Vis. Enc. | VQAv2    | TVQA     | MME      | PoPE     |
> | ------------- | ------- | --------- | -------- | -------- | -------- | -------- |
> | LLaVA-1.5     | 336     | Fixed     | 78.5     | 58.6     | 1510     | 85.9     |
> | **LLaVA-1.5** | **336** | **Tuned** | **80.4** | **59.4** | **1461** | **86.2** |
> | LLaVA-1.5     | 448     | Fixed     | 79.3     | 58.9     | 1480     | 86.7     |
> | **LLaVA-1.5** | **448** | **Tuned** | **81.1** | **62.1** | **1493** | **87.2** |
>
> ------
>
>
>
> > **Comment#4： LLaVA-HR is proposed to process input resolution of 1024, what if input images larger than 1024. Is there any extended experiments for even larger images such as 4K ones.**
>
>
>
> **Response:**  Thanks for this professional comment.   Based on your suggestion, we consider two ways to further improve resolution for  LLaVA-HR:
>
> - Resize: Directly resizing image to a larger resolution.
> - DyRes: Combining with the dynamic high resolution from LLaVA-NeXT, and adopting our dual-pathway design to each dynamic patch.
>
> As shown in the table below, the resolution of 1024 can already achieve promising results for most multimodal tasks.  Moreover, LLaVA-HR can be seamlessly combined with the dynamic high-resolution strategy of LLaVA-NeXT to further boost performance on OCR-related tasks, i.e., +3% on TextVQA.
>
> | Model           | Res  | VQAv2 | TVQA | MME  | PoPE |
> | --------------- | ---- | ----- | ---- | ---- | ---- |
> | LLaVA-HR        | 1024 | 81.9  | 67.1 | 1554 | 87.6 |
> | LLaVA-NeXT      | 1344 | 81.8  | 64.9 | 1519 | 86.5 |
> | LLaVA-HR+Resize | 1536 | 81.8  | 67.9 | 1493 | 87.7 |
> | LLaVA-HR+DyRes  | 3072 | 81.9  | 70.9 | 1450 | 88.0 |
>
> ------

---

> > ### Author Response · Authors · 2024-11-22
> >
> > > **Comment#5： What do you mean by "stages" in vision transformers?**
> >
> >
> >
> > **Response:**   We evenly divide the ViT layer into four stages, each of which receives the features of ConvNeXt through MR-Adapter. Based on your comment, we will add more explanations to our final revision.
> >
> >
> >
> > ------
> >
> > > **Comment#6： And, currently only final features from ConvNext is utilized, is there any experiments of multi-stage feature integration for that of CNN encoder?**
> >
> > **Response:**  Yes, we have tried to fuse three-stage features of  ConvNext  into ViT, but performance gains are minor. To keep the simplicity of our design, we decide to use the final features of ConvNext  in our experiments.
> >
> > | Res  | CNN Features | VQAv2 | TVQA | MME  | PoPE |
> > | ---- | ------------ | ----- | ---- | ---- | ---- |
> > | 768  | Final stage  | 81.8  | 64.3 | 1524 | 88.0 |
> > | 768  | Three stages | 81.8  | 64.6 | 1480 | 88.1 |
> >
> > ------

---

> > > ### Author Response · Authors · 2024-11-25
> > >
> > > Dear reviewer TWQ4,
> > >
> > > Thanks again for your valuable time and insightful comments. As the deadline for the Author/Reviewer discussion is approaching, it would be nice of you to let us know whether our answers have solved your concerns so that we can better improve our work. We are happy to provide any additional clarifications that you may need.
> > >
> > > Best regards!

---

> > ### Comment · Reviewer_TWQ4 · 2024-11-25
> >
> > Thanks for detailed reply. Most of my concerns have been resolved, and I recognize this work as a good one if the confusing graphs / words are modified or corrected. Thus, I think I will maintain my rating.

---

> > > ### Author Response · Authors · 2024-11-25
> > >
> > > Thanks for your encouraging comment. We would like to appreciate again for your valuable suggestions.

---

### Meta-Review · Area_Chair_3G36 · 2024-12-21

**Metareview:**

This paper proposes a mixture-of-resolution adaption method for multimodal large language model (MLLM). It consists of two visual pathways for images of different resolutions, ViT/ConveNext for low/high-resolution. The information from high-res input is adapted to low-res features by using a design of mixture-of-resolution adapter, which won't increase the token length when feeding to the LLMs, thus marginal computational overhead when using high-res image input. The experiments have shown the effectiveness and efficiency of the proposed method, and better performance than other MLLMs.

The contributions include: 1) the method is somehow novel; 2) the investigated problem is important; 3) the paper is well written and easy to understand; 4) the experimental results look good. The main concerns are 1) limited novelty (Mfi4); 2) incremental improvement on some datasets; 3) limited generation on model architecture; 4) missing additional ablations/clarification. Although most of them are well resolved by the rebuttal, some are left, including the novelty given existing work, and generalization to other MLLM architecture (only on LLaVA series). The AC also has the same concerns on these. In addition, the AC is quite concerned on why using heterogeneous vision encoders instead of homogeneous ones, if the main motivation is on mixture of resolution. Resolution is one of fundamental problems in computer vision, but this paper doesn't seem to dive deep enough. For example, ViT is not scale invariant, and simply increasing the resolution to too large one will definitely decreases the performance. Those naive resizing experiments without looking into ViT pretraining don't make much sense to me. In addition, the AC found many of the experimental details are missing. For example, what's "Resamper" in Table 1; why Resamper produces 64 tokens but compared with MRA with 576 tokens; how do you "+ConvNeXT" in the response to b6Tu's comment #1? Without the details, it is difficult to understand the values of those numbers.

However, most of the reviewers are happy with this submission and rebuttal. The AC is OK to accept it. But the authors are strongly recommended to fix those issues and make the paper more clear.

**Additional Comments On Reviewer Discussion:**

TWQ4 asked for some clarification and experiments, and was happy with the rebuttal and no change on the score.
For b6Tu, the main concerns are comparison fairness by adding ConvNext visual encoder; missing details; overall contribution is not enough. They were resolved by the rebuttal and the reviewer increased the score to 6.
For Mfi4, the main concerns are incremental improvement; limited novelty; limited generalization; missing clarification; missing experiments. Although Mfi4 didn't check in after the rebuttal, the AC thinks the generalization concern is not fully resolved, because the additional experiments are still on LLaVA series.
HwDF was concerned on small improvement on VQAv2 and missing some visualization/clarification, and happy in the end, and maintain original score.
Yq9F was concerned on missing experiments; OCRVQA result; missing clarification, but happy with the rebuttal and maintained original score.

---

### Decision · Program_Chairs · 2025-01-22

Accept (Poster)